# CENP-A and CENP-B collaborate to create an open centromeric chromatin state

Harsh Nagpal[1,6], Ahmad Ali-Ahmad [2,6], Yasuhiro Hirano [3], Wei Cai [1], Mario Halic [4], Tatsuo Fukagawa [3], Nikolina Sekulić [2,5] ✉ & Beat Fierz [1] ✉

Centromeres are epigenetically defined via the presence of the histone H3 variant CENP-A. Contacting CENP-A nucleosomes, the constitutive centromere associated network (CCAN) and the kinetochore assemble, connecting the centromere to spindle microtubules during cell division. The DNA-binding centromeric protein CENP-B is involved in maintaining centromere stability and, together with CENP-A, shapes the centromeric chromatin state. The nanoscale organization of centromeric chromatin is not well understood. Here, we use single-molecule fluorescence and cryoelectron microscopy (cryoEM) to show that CENP-A incorporation establishes a dynamic and open chromatin state. The increased dynamics of CENP-A chromatin create an opening for CENP-B DNA access. In turn, bound CENP-B further opens the chromatin fiber structure and induces nucleosomal DNA unwrapping. Finally, removal of CENP-A increases CENP-B mobility in cells. Together, our studies show that the two centromere-specific proteins collaborate to reshape chromatin structure, enabling the binding of centromeric factors and establishing a centromeric chromatin state.

Chromatin, the nucleoprotein complex composed of histone proteins and genomic DNA, is an essential regulator of processes such as gene expression and DNA replication, as it controls both local and gene-scale genome organization and accessibility[1–3]. The chromatin state is defined by local deposition of histone protein variants[4], as well as histone post-translational modifications, and by the local accumulation of a plethora of different chromatin factors[5].

CENP-A, the centromeric histone H3 variant acts as the epigenetic marker for centromere formation and subsequent DNA segregation[6–8]. CENP-A-containing nucleosomes are deposited on AT-rich DNA[9,10] within centromeric chromatin (centro-chromatin), which consists of tandem repeats of alpha-satellite DNA in human cells[11]. Within these centromeric regions, CENP-A establishes the centromere that recruits the kinetochore[12–14], a large protein complex required to bridge the chromosomes with the spindle microtubules. CENP-A exhibits a

number of structural differences compared to canonical H3[15]. A loop region in CENP-A protrudes from the nucleosome, exposing residues arginine 80 and glycine 81 (RG-loop) and thereby creating a site for CENP-N recruitment[16–18], albeit this role of the RG-loop has been challenged by recent cryoEM structures of the inner centromere[19,20] (Fig. 1a, b). Another key difference is that the αN helix of CENP-A is one turn shorter than in canonical H3, reducing DNA contacts and thus increasing the flexibility of DNA at the nucleosome entry and exit sites[21–25] (Fig. 1a, b). In accordance, cryoEM structures show that both yeast CENP-A/Cse4 and human CENP-A nucleosomes exhibit unwrapping of terminal nucleosomal DNA when bound to core kinetochore components[19,26] as well as untwisting of human CENP-A nucleosomal DNA in the context of trinucleosomes[25] when flanked by canonical H3-containing nucleosomes. This flexibility has been shown to be essential for proper chromosome segregation and the recruitment of CENP-B

[1]École Polytechnique Fédérale de Lausanne (EPFL), SB ISIC LCBM, Station 6, CH-1015 Lausanne, Switzerland. [2]Centre for Molecular Medicine Norway (NCMM), Nordic EMBL Partnership, Faculty of Medicine, University of Oslo, Oslo 0318, Norway. [3]Graduate School of Frontier Biosciences, Osaka University, Suita 565-0871, Japan. [4]Structural Biology, St. Jude Children's Research Hospital, Memphis, TN 38105-3678, USA. [5]Department of Chemistry, University of Oslo, P.O. Box 1033, Blindern 0315, Norway. [6]These authors contributed equally: Harsh Nagpal, Ahmad Ali-Ahmad. ✉e-mail: nikolina.sekulic@ncmm.uio.no; beat.fierz@epfl.ch

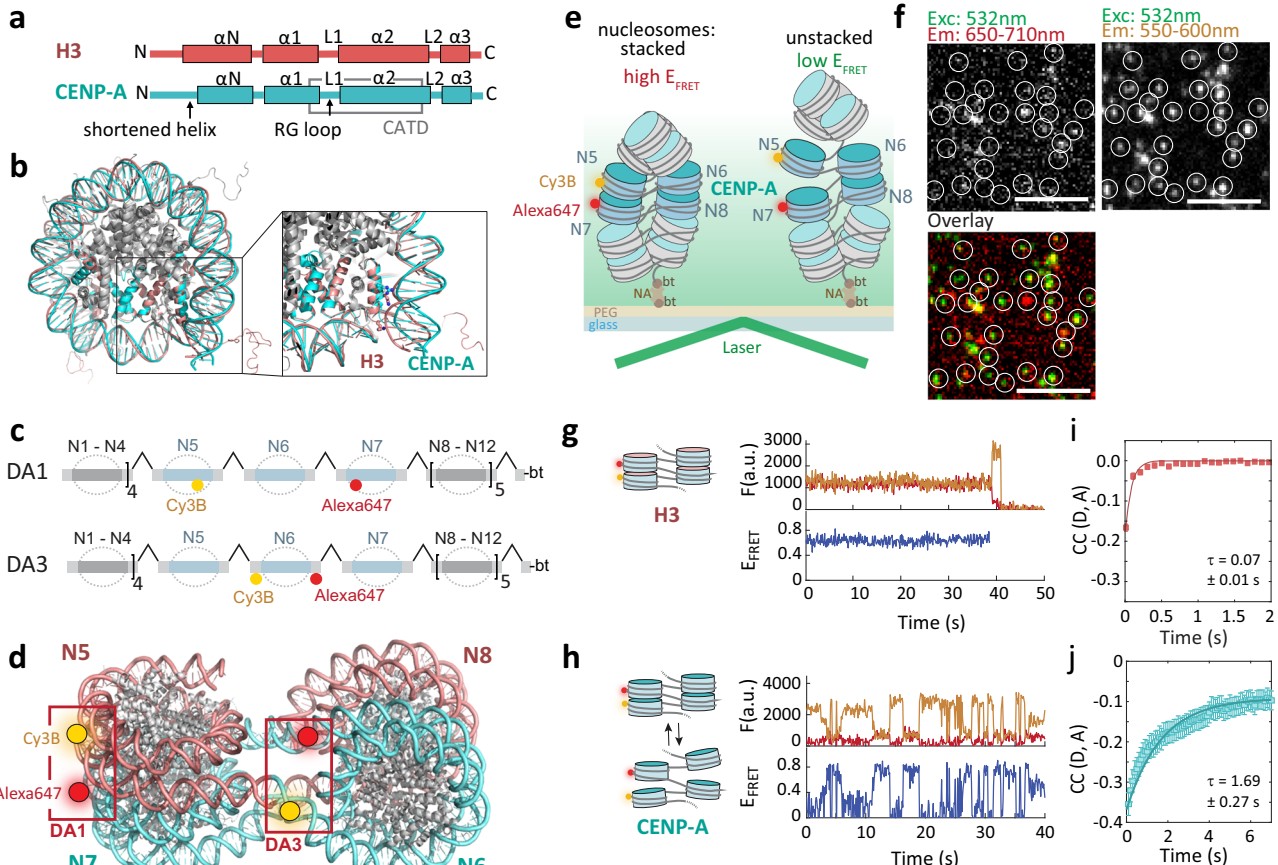

**Fig. 1 | CENP-A has a dynamic chromatin structure. a** Comparison of secondary structure organization of H3 and CENP-A. L1,2: loop 1,2; α1-3: helical segments 1-3; CATD: CENP-A targeting domain. **b** Differences in H3 and CENP-A nucleosomes, showing reduced DNA contacts of CENP-A resulting in more dynamic DNA. (PDB code: 1KX5 (H3, red), 6SE0 (CENP-A, cyan)). **c** Scheme of chromatin DNA template containing FRET donor (Cy3B, yellow) and acceptor (Alexa Fluor 647, [Alexa647], red) at nucleosome N5 and N7 (DA1, top) and at N6 (DA3, bottom). All nucleosome positioning sequences are separated by 30 bp linker DNA. **d** Tetranucleosome structure (PDB code: 1ZBB) showing stacked nucleosomes in a two-start organization. DA1 and DA3 FRET dye pairs are indicated. **e** Scheme of smFRET – TIRF experiment (DA1 illustrated). bt: biotin, NA: neutravidin. **f** Microscopic images showing FRET data of individual CENP-A-containing chromatin array at 4 mM Mg²⁺, white circles indicate the positions of immobilized fluorescently tagged chromatin, scale bar: 5 μm. For a quantification of FRET traces, refer to Fig. 2. **g** Fluorescence and FRET traces of H3 chromatin, carrying DA1 labels at 4 mM Mg²⁺, showing donor dye (orange), acceptor dye (dark red) emissions, and calculated $E_{FRET}$ trace (blue). **h** CENP-A chromatin fibers, carrying DA1 labels at 4 mM Mg²⁺, showing donor dye (orange), acceptor dye (dark red) emissions, and calculated $E_{FRET}$ trace (blue). **i** Donor–acceptor channel cross-correlation analysis of DA1-labeled H3 chromatin fibers, taken from ref. 44, Fit: cross-correlation relaxation time τ = 0.07 ± 0.01 s. Error bars: SEM. **j** Donor–acceptor channel cross-correlation analysis of DA1-labeled CENP-A (right) chromatin fibers. Fit: cross-correlation relaxation time τ = 1.69 ± 0.27 s. Traces were gated for $E_{FRET} > 0.2$, avoiding the population of zero-$E_{FRET}$ traces (see also histograms in Fig. 2a). Number of traces: n = 789 (H3), n = 485 (CENP-A). Error bars: SEM. Source data are provided as a Source Data file.

and CENP-C[22]. However, the question of how the increased flexibility of CENP-A nucleosomes alters chromatin structure remains unclear.

The constitutive centromeric protein CENP-B is the only component of centromeric chromatin to make sequence-specific DNA contacts[27]. It recognizes a defined DNA sequence, the CENP-B box motif (B-Box) within alpha-satellite repeats[28–30] via its N-terminal DNA-binding domain. While CENP-B is nonessential for centromere function[31–33], it has been shown to be required for de novo centromere assembly in human artificial chromosomes (HACs)[34–36], as well as to stabilize the kinetochore and enhance centromere function[37,38]. Indeed, depletion of CENP-B greatly increases cell lethality upon CENP-A loss[39]. Finally, CENP-B acts as an anchor for diverse chromatin proteins, including heterochromatin protein 1 (HP1) and the H3K36-specific methyltransferase ASH1L[40]. These combined functions render CENP-B a key regulator of the centromeric chromatin state, and consequently, an important factor controlling the structure and stability of the centromere.

B-Boxes are regularly distributed in alpha-satellite DNA, occurring generally within every second repeat and being localized near the DNA entry-exit region of centromeric nucleosomes in humans[41]. Binding these DNA sequences requires CENP-B to invade centromeric chromatin structure. CENP-B localization is not limited to CENP-A nucleosomes[28]; however, previous studies implied that its DNA-binding domain (DBD) can make preferential contacts to CENP-A through interaction with the amino tail of CENP-A[37,38]. Moreover, CENP-B is able to homodimerize through its C-terminal domain and bridge B-Boxes in two alpha-satellite monomers[42], forming chromatin loops[43]. The possibility for CENP-B to invade centro-chromatin structure and to cross-bridge chromatin elements requires significant flexibility of centromeric chromatin. How CENP-B remodels centromeric chromatin structure is, however, an unsolved question.

We have recently developed single-molecule fluorescence resonance energy transfer (smFRET) methods to reveal dynamic chromatin organization[44]. These studies revealed that chromatin exhibits micro- to millisecond conformational fluctuations, which transiently expose internal DNA segments. Such structural fluctuations are exploited by DNA-binding proteins, e.g., transcription factors, to access their binding sites within compact chromatin[45]. Here, we employ single-molecule fluorescence methods, including smFRET and single-molecule colocalization imaging, to probe the dynamic structure of

centro-chromatin. Using reconstituted CENP-A chromatin, we find that centromeric histones induce a highly dynamic chromatin state, which enables unhindered DNA access to CENP-B. Using cryoEM analysis and smFRET we further show that CENP-B binding induces chromatin opening by DNA unwrapping and distorting linker DNA containing its B-Box binding sequences. Finally, we find that the two centromeric proteins dynamically collaborate in cells, where the presence of CENP-A stabilizes CENP-B retention at centromeres. In conclusion, we demonstrate a chromatin-mediated crosstalk between two key centromeric proteins, establishing a dynamic and accessible chromatin state at centromeres.

## Results

### CENP-A creates a dynamic chromatin structure

Multiple structural studies have indicated that CENP-A mononucleosomes are more flexible compared to H3 nucleosomes[21,22,25,41,46–51], although how this increased local flexibility modulates longer-range higher-order chromatin structure is unclear. Increased nucleosomal flexibility might lead to a more dynamic higher-order structure. Indeed, cryo-EM analysis of CENP-A containing tri-nucleosomes suggested a highly heterogeneous untwisted structure[25]. Conversely, CENP-A might allow for the formation of compact structures by relieving conformational strain. In accordance with this hypothesis, sedimentation assays previously showed that CENP-A increased the formation of compacted chromatin structures[52,53].

We decided to probe the dynamic organization of CENP-A-containing centro-chromatin by employing a recently established single-molecule fluorescence resonance energy transfer (smFRET) method, measuring energy transfer between two fluorescent dyes placed at defined positions within chromatin fibers and reporting on chromatin structural parameters[44,54]. Using this approach and varying FRET donor and acceptor dye placement, we can directly detect nucleosome stacking or determine the orientation of linker DNA on a single-fiber level. To implement the approach, we reconstituted 12-mer chromatin fibers using histone octamers containing either CENP-A or H3, as well as all other canonical human histone proteins, namely H4, H2A and H2B.

The DNA template used for chromatin fiber assembly contained 12 tandem repeats of the 601 nucleosome positioning sequence (NPS)[55], ensuring nucleosome placement with base pair accuracy, which is instrumental for smFRET measurements. In contrast, constructs containing natural alpha-satellite repeats did not result in sufficiently well-defined nucleosomes for reliable smFRET experiments, in particular in combination with CENP-A octamers. Individual 601 NPSs were further separated by 30 bp linker DNA (Fig. 1c). FRET donor (Cy3B) and acceptor dyes (Alexa647) were placed on nucleosome 5 (N5) and 7 (N7) (called the donor-acceptor 1 (DA1) pair[44], Fig. 1d) using a DNA plug-and-play strategy[56] (see Supplementary Fig. 1 for analytical data of reconstitutions and Supplementary Tables 1–3 for all DNA constructs). This FRET pair allows the detection of nucleosome stacking, i.e., face-to-face contact between next-neighbor nucleosomes (Fig. 1d). To probe the relative orientation of linker DNA, we further used alternative FRET dye positions, named DA3[44], where the dyes flank the central nucleosome N6 (Fig. 1c, d, Supplementary Fig. 1, Supplementary Tables 1–3). Together, these two labeling schemes enable us to probe both DNA orientation and nucleosome stacking within a central tetranucleosome unit in the chromatin fiber.

Reconstituted chromatin fibers, either fully occupied by H3 or by CENP-A containing nucleosomes, hereon called H3 chromatin or CENP-A chromatin, respectively, were then immobilized in a flow channel, followed by single-molecule fluorescence imaging via total internal reflection fluorescence (smTIRF) microscopy (Fig. 1e). Individual chromatin fibers were detected as diffraction-limited spots in the donor- and acceptor-emission channels (Fig. 1f). We recorded donor and acceptor dye fluorescence emission over time, and calculated time-traces of FRET efficiency ($E_{FRET}$) (Fig. 1g, Methods). For H3 chromatin fibers at 4 mM $Mg^{2+}$, conditions that induce chromatin compaction via shielding of negative DNA charges[57], this resulted in stable $E_{FRET} \sim 0.5$ over several tens of seconds[44] (Fig. 1g). This is in agreement with previous experiments that used confocal single-molecule fluorescence spectroscopy of H3-containing chromatin to reveal rapid structural fluctuations on the μs–ms timescale[44], which are mostly undetectable on the slower smTIRF timescale. In contrast, when observing CENP-A chromatin under the same conditions, we observed large-scale fluctuations in FRET efficiency on the seconds timescale for a substantial fraction of CENP-A-containing chromatin fibers (Fig. 1h). Donor-acceptor cross-correlation analysis of these fluctuations, including all traces that showed an average $E_{FRET} > 0.2$, revealed only small-amplitude motions with $\tau_R \sim 0.1$ s for H3 chromatin[44] (Fig. 1i). In comparison, CENP-A chromatin exhibited large-amplitude motions with a relaxation time $\tau_R = 1.69 \pm 0.27$ s (Fig. 1j). Together, these results show that CENP-A renders chromatin fibers to be much more dynamic, as opposed to the more stably folded H3 chromatin. Of note, in human centromeres, not all centromeric repeats are occupied by CENP-A[58]. Here, we monitor structure over three consecutive CENP-A nucleosomes. This thus represents a region of high local CENP-A content. Other regions of the centromere, containing substantial amounts of H3 nucleosomes, are expected to exhibit intermediate dynamics between CENP-A and H3 chromatin.

### CENP-A disrupts nucleosome-stacking interactions

Comparing the distribution of $E_{FRET}$ values between H3 and CENP-A chromatin provided further insights into the organization of centromeric chromatin. We constructed histograms of $E_{FRET}$ values, integrating all time traces. The resulting distributions could be described by the sum of two Gaussian distributions, corresponding to a low $E_{FRET}$ (open) and mid-to-high $E_{FRET}$ (compacted) conformation (Fig. 2a, Supplementary Table 4).

First, we focused on chromatin fibers containing the DA1 dye pair, monitoring stacking interactions between next-neighbor nucleosomes (N5-N7). At low ionic strength (40 mM KCl) and in the absence of bivalent cations (0 mM $Mg^{2+}$), we observed that CENP-A chromatin was essentially open with a majority of molecules showing low FRET ($E_{FRET} < 0.1$, Fig. 2a). H3 chromatin, in contrast, exhibited a majority population of intermediate FRET efficiency ($E_{FRET} = 0.2$–0.5, Fig. 2b). This is consistent with the existence of rapidly exchanging populations of open or stacked nucleosomes, with exchange kinetics on the μs–ms timescale[44]. Addition of bivalent cations (4 mM $Mg^{2+}$) induced chromatin compaction in both CENP-A and H3 chromatin, as exhibited by a high-FRET population centered at $E_{FRET}$ values of 0.44 for CENP-A and 0.57 for H3 chromatin (Fig. 2a, b). This population was markedly smaller for CENP-A, and characterized by a broad distribution of $E_{FRET}$ values due to large-scale conformational dynamics in the seconds timescale (Fig. 1h and Fig. 2a). We then quantified the number of single-molecule traces, which (i) showed large-scale fluctuations as in Fig. 2a (labeled as "Dynamic"), (ii) showed stable medium-to-high $E_{FRET}$ values (labeled as "High FRET") or (iii) exhibited low $E_{FRET}$ values (labeled as "Low FRET") for both CENP-A and H3 chromatin (Fig. 2c). Also, in this analysis, CENP-A chromatin showed a much lower percentage of traces at high $E_{FRET}$ compared to H3 chromatin, but a significantly increased percentage of dynamic traces, corroborating its open and dynamic organization.

Alternatively, detecting FRET via the DA3 FRET pair provides information on the orientation of the linker DNA that flanks the central nucleosome within the reconstituted fibers (N6, Fig. 1d). In CENP-A chromatin, we observed two populations of low- and mid-FRET ($E_{FRET} = 0.18$) values at low ionic strength, indicating that the DNA in CENP-A nucleosomes is divergent at the entry-exit sites, even in a chromatin fiber context. Upon the addition of 4 mM $Mg^{2+}$, a higher FRET state was populated ($E_{FRET} = 0.32$), indicating transient higher-

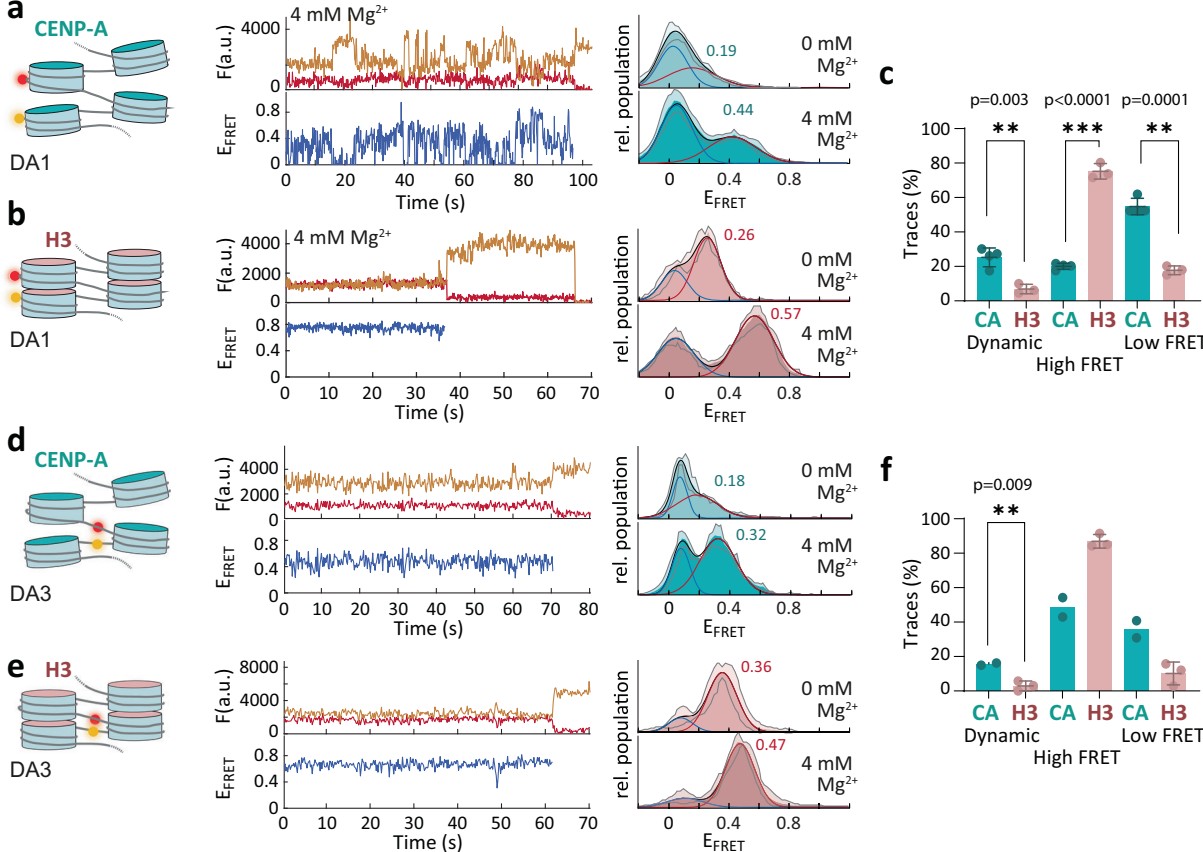

**Fig. 2 | CENP-A generates open chromatin. a** Left: Scheme of tetranucleosome within CENP-A chromatin fiber with DA1 arrangement of donor (orange) and acceptor (red) FRET dyes. Middle: Single-molecule traces (donor: orange, acceptor: red, FRET: blue) for CENP-A chromatin, labeled at DA1, at 0 mM Mg$^{2+}$ (top), 4 mM Mg$^{2+}$ (bottom) until either donor or acceptor dye were photobleached. Right: FRET population histograms, observed for CENP-A chromatin, labeled at DA1, at the indicated Mg$^{2+}$ concentrations. All Histograms are the average of n = 4 independent repeats. Histograms are fitted with Gaussian functions (red). Peak E$_{FRET}$ for high FRET populations are indicated. Shaded region shows S.D. **b** Scheme and single-molecule traces for H3 chromatin, labeled at DA1, and accompanying FRET histograms at the indicated Mg$^{2+}$ concentrations. All Histograms are the average of $n = 4$ (0 mM MgCl$_2$), $n = 3$ (4 mM MgCl$_2$) independent repeats. Shaded region shows S.D. **c** Percentage of dynamic, high-FRET, and low-FRET traces of CENP-A (CA) and H3

chromatin for DA1. $n = 4$ (CA), $n = 3$ (H3). Error bars show mean value +/− S.D. **\*\***$P < 0.01$,**\*\*\***$P < 0.001$ using 2-tailed unpaired $t$-test. **d** Scheme and single-molecule traces for CENP-A chromatin, labeled at DA3, and accompanying FRET histograms at the indicated Mg$^{2+}$ concentrations. All Histograms are the average of $n = 2$ independent repeats. Shaded region shows S.D. **e** Scheme and single-molecule traces for H3 chromatin, labeled at DA3, and accompanying FRET histograms at the indicated Mg$^{2+}$ concentrations. All Histograms are the average of $n = 3$ independent repeats. Shaded region shows S.D. **f** Percentage of dynamic, high-FRET, and low-FRET traces of CENP-A (CA) and H3 chromatin for DA3. $n = 2$ (CA), $n = 3$ (H3). Error bars show mean value +/− S.D. **\*\***$P < 0.01$ using 2-tailed unpaired $t$-test. **a**–**e** For all fit values and detailed statistical information (number of traces, replicates), see Supplementary Table 4. Source data are provided as a Source Data file.

order structure stabilization. Conversely, the distance between DNA strands at the nucleosome entry-exit sites was smaller in H3 chromatin (Fig. 2e), as observed by higher FRET values for both low ionic strength (E$_{FRET}$ = 0.32) and at 4 mM Mg$^{2+}$ (E$_{FRET}$ = 0.47). From the DA3 vantage point which monitors the relative position of the linker DNA, dynamics were less pronounced but still more prevalent in CENP-A chromatin compared to H3 chromatin (Fig. 2f). To gain further insights into the conformation of CENP-A chromatin, we compared measured E$_{FRET}$ to calculated E$_{FRET}$ values that were generated using cryo-EM structures of H3-H3-H3 or H3-CENP-A-H3 trinucleosomes[25] (Supplementary Fig. 2). Our measurements for H3 were compatible with expected values from the cryo-EM structure. In contrast, our measured E$_{FRET}$ values were higher than those expected from the structure of a H3-CENP-A-H3 trinucleosome exhibiting an untwisted central nucleosome. Our results thus indicate that fiber interactions constrain CENP-A nucleosomes into a more H3-like conformation in an extended fiber context (Supplementary Fig. 2).

Together, our data show that CENP-A nucleosomes can engage in stacking interactions, forming tetranucleosome units, albeit at lower efficiency and in a distorted architecture. Moreover, CENP-A

chromatin exhibits greatly increased dynamics associated with large-scale breathing motions on the seconds timescale. Both the distorted structure and dynamic openings may result in increased local DNA access within centromeric regions.

## CENP-B efficiently binds nucleosomes independent of histone type

We thus wondered if CENP-A chromatin incorporation may provide a particularly permissive chromatin state for DNA-binding proteins. CENP-B is a DNA-binding factor of particular relevance, as it is a component of centromeric chromatin and is instrumental for the establishment of a centro-chromatin state[43]. To determine how CENP-A controls DNA access for CENP-B, we directly determined the binding dynamics of full-length CENP-B using single-molecule colocalization imaging[45,59]. In such an experiment, individual naked DNA molecules or reconstituted nucleosomes, all labeled with a far-red fluorescent dye are surface-immobilized within a microfluidic flow-channel (Fig. 3a). Their individual positions are then detected via single-molecule total internal reflection fluorescence (smTIRF) microscopy. Subsequently, recombinant full-length CENP-B, which is labeled by a fluorescent dye

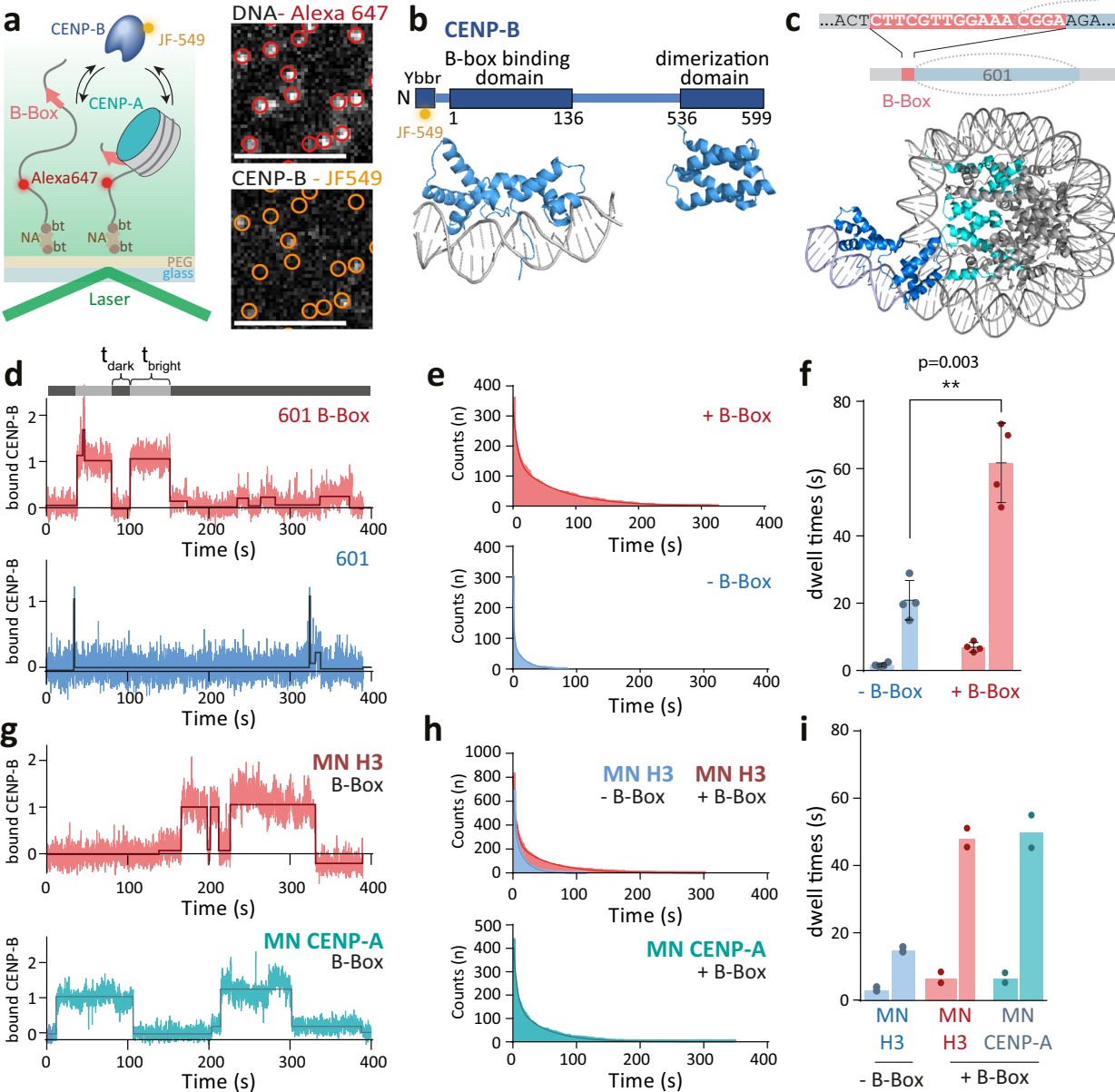

**Fig. 3 | CENP-B binds independently of histone type. a** Left: Scheme of smTIRF experiment to detect CENP-B binding to B-Box- or non-B-Box-containing 601 DNA or nucleosomes. Right: smTIRF image showing immobilized 601-B-Box DNA in the far-red channel (top, red circles) and CENP-B binding in the green-orange channel (bottom). Scale bar: 5 μm. **b** Used CENP-B construct. Ybbr: peptide tag for fluorescent labeling. Below: Structures of the DNA-binding domain (PDB code: 1HLV) and dimerization domain (PDB code: 1UFI). **c** Top: Position of the B-Box DNA sequence relative to the CENP-A nucleosome. Bottom: Model of a CENP-A nucleosome containing a B-Box-bound CENP-B. **d** Fluorescence time trace (blue) of CENP-B binding events to 601-B-Box (top) and non-B-Box DNA (bottom). Free ($t_{dark}$) and bound times ($t_{bright}$) are determined via thresholding. **e** Cumulative histogram of CENP-B binding to B-Box- (red) and non-B-Box- (blue) containing DNA, fitted by a

bi-exponential function (solid line). **f** Dissociation time constants ($t_{off,1}$ and $t_{off,2}$) of CENP-B to non-B-Box (blue) and B-Box (red) 601 DNA. Width of bars indicates relative amplitudes $A_1$ and $A_2$. For all fit parameters, see Supplementary Table 5. $n = 4$ independent repeats. Error bars show mean value +/− S.D. \*\*$P < 0.01$, 2-tailed unpaired $t$-test. **g** Representative fluorescence time trace (blue) of CENP-B binding events to 601-B-Box nucleosomes (MN) containing H3 (top) or CENP-A (bottom). **h** Cumulative histogram of CENP-B binding to B-Box nucleosomes, containing H3 (top) or CENP-A (bottom) fitted by a bi-exponential function (solid line). **i** Dissociation time constants of CENP-B to H3 non-B-Box (blue), H3 B-Box (red) and CENP-A B-Box (gray) 601 nucleosomes. $n = 2$ independent repeats; Source data are provided as a Source Data file.

in the green-orange spectrum is injected into the channel, and individual dynamic binding events are detected by colocalization imaging between the far-red (DNA/chromatin) and orange (CENP-B) channels.

To implement this experiment, we prepared recombinant full-length CENP-B using a DNA construct that contained both the CENP-B DNA-binding and dimerization domains and carrying an N-terminal peptide tag that allowed chemoenzymatic labeling[60] with a high-performance green-orange dye (JF-549[61], Fig. 3b and Supplementary Fig. 3). Overall labeling efficiency was around 50%, ensuring that most

CENP-B dimers carried one fluorophore. In parallel, we generated a DNA template composed of a 601 NPS, either without any sequence changes (601) or containing a CENP-B Box (601 B-Box) that was positioned 69 bp from the 601 dyad position, similar to previous studies[38]. This position mimics the location of the B-Box relative to nucleosomes within human alphoid DNA[41] (Fig. 3c, Supplementary Fig. 3 and Supplementary Tables 1–3). The DNA was further modified with a far-red fluorescent dye (Alexa Fluor 647) and a biotin moiety for immobilization.

We proceeded to measure CENP-B binding dynamics to immobilized naked DNA, with or without a B-Box, via single-molecule colocalization imaging. For these experiments, we injected CENP-B at a concentration chosen such that we were able to clearly observe single CENP-B binding events (approx. 5 nM) and under native ionic strength (150 mM salt) (Fig. 3a, d). For each detected DNA molecule in the far-red channel, we then generated time traces of CENP-B binding observed in the green-orange channel, which allowed us to determine the length of individual binding events ($t_{bright}$) and the time between binding interactions ($t_{dark}$, Fig. 3d). Lifetime histograms of binding events, generated from $t_{bright}$, provided information about CENP-B residence times, whereas histograms of $t_{dark}$ reported on binding rates (Supplementary Fig. 4). Focusing on residence times, we analyzed the lifetime histograms of the bound state using a bi-exponential fit, resulting in a short ($\tau_{off,1}$) and long ($\tau_{off,2}$) residence times (Fig. 3e, for all fit parameters, see Supplementary Table 5). This is commonly observed for chromatin binding proteins[45,59,62], and indicates different interaction modes, usually interpreted as non-specific charge-based interactions ($\tau_{off,1}$) and engagement of specific DNA sequences via the DBD ($\tau_{off,2}$).

In the presence of a B-Box, around half of all CENP-B-DNA interaction exhibited a short residence time ($\tau_{off,1} = 6.9 \pm 1.4$ s), whereas the remainder showed a long residence time ($\tau_{off,2} = 61 \pm 11$ s, Fig. 3f). This is consistent with a mid-nanomolar affinity, as determined by electrophoretic mobility shift assays (Supplementary Fig. 5), and also with previous results[43]. These long, multi-second residence times, comparable to those of transcription factors[45,63,64], enable CENP-B to play a role in chromatin structure organization, while ensuring a dynamic chromatin state. Importantly, full-length CENP-B was required for stable DNA binding: residence times for a short CENP-B construct, encompassing only the DNA-binding domain (residues 1–150), were strongly reduced ($\tau_{off,1} = 2.6 \pm 0.2$ s, $\tau_{off,2} = 22.8 \pm 4.4$ s) (Supplementary Fig. 6). These findings suggest that the centromere-binding and dimerization domains enhance B-Box recognition by CENP-B. Finally, the absence of a B-Box sequence resulted in significantly shorter binding times for full-length CENP-B ($\tau_{off,1} = 1.6 \pm 0.6$ s, $\tau_{off,2} = 20 \pm 6$ s, Fig. 3f). In conclusion, CENP-B can also bind DNA in a sequence-independent fashion, possibly allowing an efficient search process for B-Box sequence elements in the genome[65].

We then tested whether CENP-B can efficiently invade and bind to nucleosomal DNA. To this end, we reconstituted either H3- or CENP-A-containing nucleosomes on the previously used DNA templates (Supplementary Fig. 3), with the CENP-B box precisely positioned at the DNA-entry-exit site as in human alphoid DNA[41]. These nucleosomes were used to measure CENP-B binding dynamics via single-molecule colocalization imaging (Fig. 3g). Interestingly, the presence of H3-containing histone octamers did not greatly affect the observed CENP-B residence times (Fig. 3g–i). CENP-B was efficiently able to invade (Supplementary Fig. 4) and remain bound at H3-containing mononucleosomes with residence times $\tau_{off,1} = 6.9 \pm 2.3$ s (66% of all events) and $\tau_{off,2} = 48 \pm 4$ s (34% of all events), similar to those for naked DNA (Fig. 3i).

Similarly, CENP-B was able to bind CENP-A-containing mononucleosomes basically unhindered, with residence times $\tau_{off,1} = 6.8 \pm 2.06$ s (51% of all events) and $\tau_{off,2} = 50 \pm 7$ s (49% of all events, Fig. 3g–i, see Supplementary Table 5 for all data). Within CENP-A nucleosomes, the peripheral DNA is accessible[21] and thus the CENP-B box is exposed, which explains a negligible effect on CENP-B binding kinetics. Conversely, we also could not detect any positive effect of CENP-A to promote the binding of full-length CENP-B, e.g., via direct interactions as previously suggested from experiments using the CENP-B DBD alone[38]. In H3-containing nucleosomes, the CENP-B box is partially located within the nucleosome structure (Fig. 3c). The fact that nucleosomes do not reduce CENP-B binding or residence times in this context therefore indicates that the interactions of CENP-B to its 17-bp DNA recognition sequence can efficiently outcompete peripheral histone-DNA contacts, independently of histone type.

## CENP-B binding is affected by the underlying chromatin state

In the cell, CENP-B does not interact with individual nucleosomes but binds to extended chromatin fibers, which can adopt various conformations and higher-order structures[1]. While our single-molecule experiments revealed that CENP-B binding is not enhanced by CENP-A in a single-nucleosome context, we hypothesized that the open and dynamic nature of centromeric chromatin may present a preferred binding environment.

We proceeded to test this hypothesis by measuring CENP-B binding kinetics to reconstituted 12-mer chromatin fibers containing either H3 or CENP-A (Fig. 4a). Here, we positioned a single B-Box site within the central nucleosome (N6) at a distance of 69 bp from the dyad (Fig. 4b), similar to our previous binding studies with mononucleosomes (Fig. 3). This DNA design allows to detect binding dynamics without confounding effects of multivalent CENP-B binding to neighboring B-Boxes. These chromatin fibers containing either H3 or CENP-A were then immobilized in flow cells to determine CENP-B binding kinetics by single-molecule colocalization imaging. Importantly, we performed these experiments under physiological salt conditions (150 mM KCl) that result in a compacted chromatin structure[45].

In H3 chromatin, the detected specific residence times $\tau_{off,2}$ for CENP-B were substantially reduced by almost 3-fold to $\tau_{off,2} = 17.6 \pm 6.7$ s compared to those in mononucleosomes (Fig. 4c–e). X-ray crystallography studies revealed that B-Box DNA is distorted in the CENP-B-bound state[30,66]. Such distortion of the linker DNA is most likely incompatible with stacked nucleosome conformations that are observed in structures of compact tetranucleosomes or chromatin fibers[25,67,68]. Thus, CENP-B may not be able to form an optimal DNA-bound complex, resulting in its rapid eviction from chromatin fiber structure.

Conversely, we did not observe such inhibition in chromatin fibers containing CENP-A nucleosomes: Here, specific residence times for CENP-B were restored to $\tau_{off,2} = 51.4 \pm 5.3$ s, similar to those in free mononucleosomes (Fig. 4c–e). This recovery of binding lifetimes within CENP-A-containing chromatin suggest that CENP-A chromatin fibers can accommodate CENP-B, tolerating bent DNA, and provide less hindered DNA access.

## CENP-B binding destabilizes DNA wrapping of nucleosomes

To explore the effect of CENP-B binding on the conformation of linker DNA within nucleosome arrays we used smFRET with linker-positioned dyes (DA3) (Fig. 5a, b). When we incubated H3 chromatin with CENP-B, the detected FRET efficiency barely changed, indicating that CENP-B was not able to efficiently invade chromatin fibers when DNA is more rigidly held in place by histone contacts (Fig. 5c). Conversely, we detected a loss of smFRET in CENP-A chromatin upon incubation with CENP-B (Fig. 5d). Thus, CENP-B is able to bind and further open the nucleosomal DNA in CENP-A arrays, where linker DNA is already more dynamic.

Next, we wanted to understand the molecular details of the CENP-B binding to the nucleosomal DNA linker. It was previously proposed[38] that CENP-B forms a more stable complex with CENP-A nucleosome and that the DNA-binding domain of CENP-B interacts directly with CENP-A, but high resolution of this interaction was never reported. To understand the molecular details of CENP-B binding in the context of CENP-A nucleosomes, we used cryoEM to analyze CENP-A nucleosomes assembled on 601 B-Box DNA with and without full length CENP-B bound (Supplementary Fig. 7). We observed subtle differences in nucleosome structure when comparing particles with and without bound CENP-B in the context of 601 DNA. In a majority of particles, it was difficult to detect a clear density for CENP-B, most probably due to

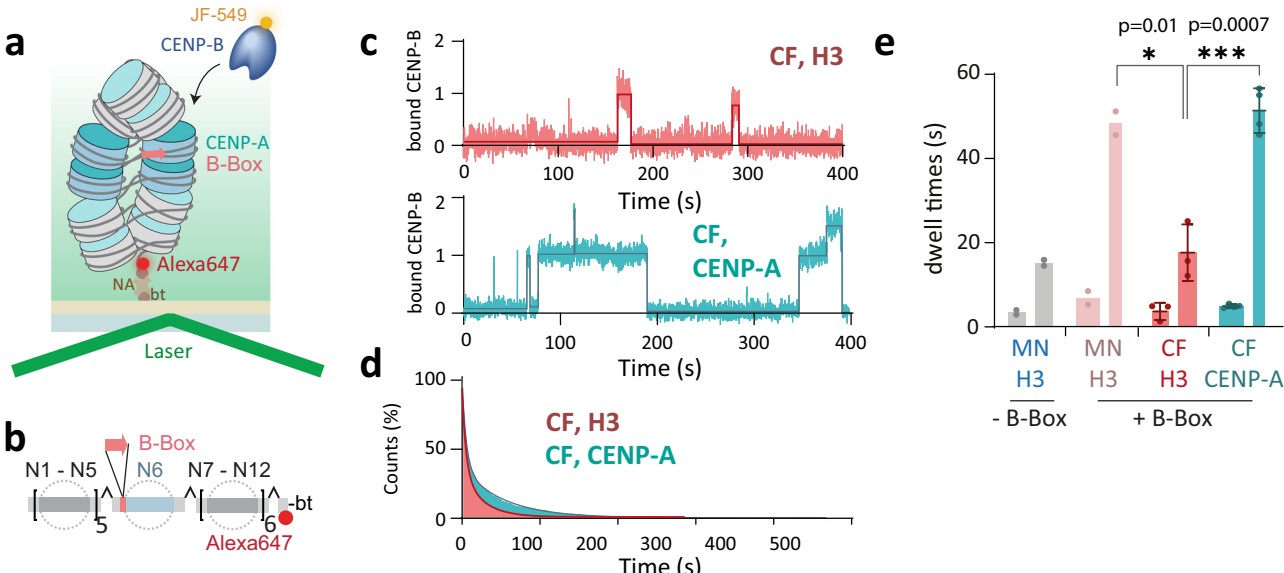

**Fig. 4 | CENP-B binding is controlled by the local chromatin state. a** Scheme of colocalization experiment to detect CENP-B binding to immobilized H3C- or CENP-A-containing chromatin fibers. **b** Scheme of DNA template containing one B-Box at nucleosome N6 (bottom). bt is biotin, NA is neutravidin. **c** Representative fluorescence time trace of CENP-B binding events to H3- (top) and CENP-A- (bottom) containing chromatin fibers (CF). The traces are fitted by a thresholding algorithm. **d** Cumulative histogram of CENP-B binding to H3 arrays (orange) and CENP-A arrays (green) fitted by a bi-exponential function (solid line). **e** Dissociation time constants ($\tau_{off,1}$ and $\tau_{off,2}$) of CENP-B to the indicated nucleosome and chromatin constructs. CF: chromatin fiber. Thickness of the bars indicate amplitudes $A_i$ associated with the indicated dwell times. $n = 2$ (MN H3, -B-Box), $n = 2$ (MN H3, +B-Box), $n = 3$ (CF H3, +B-Box), $n = 4$ (CF CENP-B, +B-Box) independent repeats. Error show mean value +/− S.D. *$P < 0.05$, **$P < 0.01$,***$P < 0.001$ using 2-tailed unpaired $t$-test. For all fit values, see Supplementary Table 5. Source data are provided as a Source Data file.

high conformational heterogeneity. In the subset of particles where CENP-B density was observed, we could not detect any direct interaction between CENP-A histone and CENP-B. Instead, in this class the DNA was lifted from the nucleosome body at the entry/exit site (Supplementary Fig. 7f–j, Movie S1). This result is consistent with the reduced FRET efficiencies observed in smFRET experiments (Fig. 5d), and also points towards an increased DNA flexibility induced by CENP-B.

In human cells, the CENP-B box is naturally present in the context of AT-rich alpha-satellite DNA[28,69] and we know that DNA sequence plays a role in DNA wrapping[15,70] and nucleosome stability[71,72]. We thus extended our studies to determine the effect of CENP-B binding on nucleosomes assembled on 171 bp alpha-satellite DNA from the human X chromosome containing a functional CENP-B box[41,73]. First, we investigated the distance between DNA ends using an ensemble FRET assay. To this end, we attached Cy3 and Cy5 fluorophores at the end of alpha-satellite DNA (Fig. 5e and Supplementary Tables 1–2) and assembled H3 and CENP-A mononucleosomes (H3$^{\alpha\text{-sat}}$ and CENP-A$^{\alpha\text{-sat}}$, respectively). Mononucleosomes were then incubated with an increasing concentration of FL CENP-B and FRET was measured between the fluorophores. FRET between DNA ends on H3 nucleosomes in the absence of CENP-B is higher than on CENP-A nucleosomes (Fig. 5e, f), indicating more flexible DNA ends on CENP-A nucleosomes, consistent with previous studies on mononucleosomes[21,22,25,41,46–50] and with ours and others[52] observations in nucleosome arrays (Fig. 5c, d). Binding of CENP-B decreases the FRET signal in both types of nucleosomes, indicating further opening of nucleosomal DNA ends.

To gain deeper insights into the effect of CENP-B on nucleosome structure in the context of alpha-satellite DNA, we resorted to single-particle cryoEM. In the absence of CENP-B, a larger fraction of H3$^{\alpha\text{-sat}}$ nucleosomes exhibited completely wrapped nucleosomal DNA (74.3% of all observed nucleosomes), compared to CENP-A$^{\alpha\text{-sat}}$ nucleosomes (with 67.2% nucleosomes fully wrapped) (Fig. 5g and Supplementary Fig. 8). Moreover, completely wrapped H3$^{\alpha\text{-sat}}$ nucleosomes showed

tighter DNA wrapping, with a distance of 70 Å between super helical location (SHL) 7 and SHL -7, in contrast to fully wrapped CENP-A$^{\alpha\text{-sat}}$ nucleosomes where the same distance measured 80–90 Å (Fig. 5g). These data are consistent with our FRET experiments indicating higher flexibility of the nucleosomal DNA ends for CENP-A nucleosomes (Fig. 5e, f). Interestingly, upon binding CENP-B, CENP-A$^{\alpha\text{-sat}}$ nucleosome showed striking nucleosomal DNA unwrapping. The nucleosomal DNA was released at both ends, with only 12.7% of the particles remaining in the wrapped state (exhibiting even larger distance of 100 Å between SHL 7 and SHL -7, comparing to nucleosomes without bound CENP-B) (Fig. 5f, g and Supplementary Fig. 8d, e). Furthermore, solely in the presence of CENP-B, we observed classes with nucleosomes unwrapped up to SHL 5 (Fig. 5f). The extremely high flexibility of nucleosomal DNA ends in the presence of CENP-B indicates DNA disorder caused by CENP-B and further supports that the lack of clear observable CENP-B density in the cryoEM data is due to conformational heterogeneity. Together, our results indicate that binding of CENP-B to CENP-A nucleosomes results in decreased order of the DNA ends rather than the stabilization of specific CENP-A/CENP-B interactions that favor open nucleosomal DNA ends. Moreover, the pronounced unwrapping we observe in the context of alpha-satellite DNA would lead to large-scale chromatin rearrangement if it persists in the context of nucleosome arrays.

In summary, combined cryoEM and FRET analyses of nucleosomes directly show that CENP-B binding leads to destabilization of terminal nucleosomal DNA and that this effect is even more pronounced in the context of alpha-satellite DNA.

## CENP-B binding pries opens chromatin structure
We thus wondered how CENP-B might affect chromatin structure beyond individual nucleosomes. Indeed, a critical function of CENP-B is its ability to cross-bridge chromatin elements and induce loops in chromatin structure, as it exists as a dimer able to bind to B-Box DNA motifs simultaneously[43]. We thus wondered if CENP-B would be able to

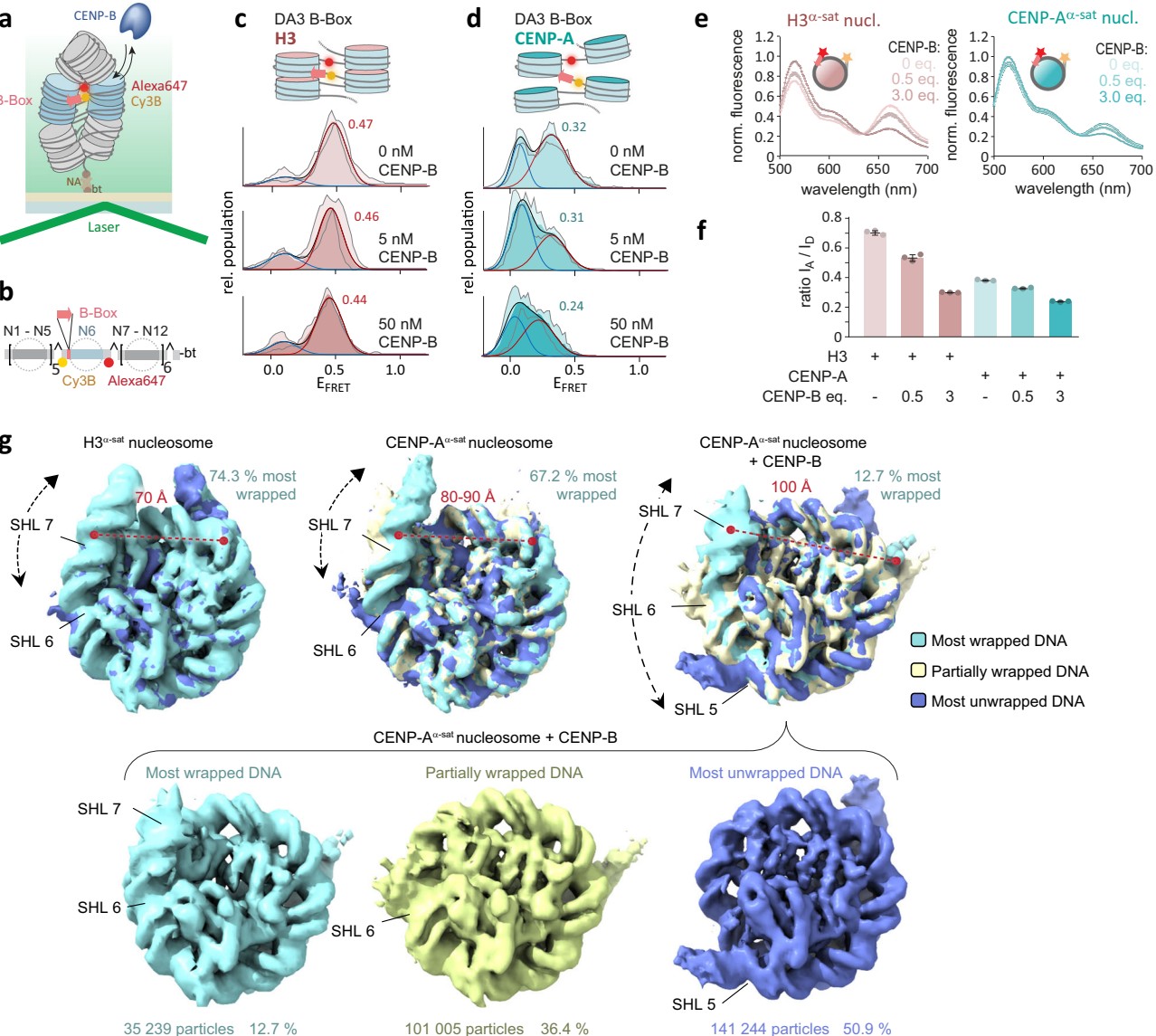

**Fig. 5 | CENP-B binding destabilizes DNA wrapping of nucleosomes. a** Scheme of smFRET experiment. **b** Scheme of chromatin DNA containing FRET donor (Cy3B, yellow) and acceptor (Alexa fluor 647, red) and 1xB-Box at N6 (DA3BB). **c** FRET histograms for H3 DA3BB at indicated CENP-B concentrations. Histograms are fitted with Gaussian functions (red), $E_{FRET}$ values of the high FRET population are displayed. All Histograms are the average of $n = 2$ independent repeats. Shaded region shows S.D. **d** CENP-A DA3 histograms at indicated CENP-B concentrations with the $E_{FRET}$ value of the high FRET population displayed. Peak $E_{FRET}$ for high FRET populations are indicated. All Histograms are the average of $n = 2$ independent repeats. Shaded region shows S.D. **e** Left: Fluorescence spectra for H3$^{\alpha\text{-sat}}$ nucleosomes upon incubation with indicated equivalents of CENP-B. Right: Fluorescence spectra for CENP-A$^{\alpha\text{-sat}}$ nucleosome upon incubation with CENP-B. The scheme shows the position of Cy3 (red) and Cy5 (yellow) fluorophores. **f** Ratio of the acceptor dye emission ($I_A$, at 662 nm) over the donor dye emission ($I_D$, at 565 nm)

for H3$^{\alpha\text{-sat}}$ and CENP-A$^{\alpha\text{-sat}}$ nucleosomes at the indicated CENP-B molar equivalents. Error bars show mean value +/− S.D., $n = 3$ independent replicates. **g** Top row, from left to right: overlay of 3D cryo-EM maps for H3$^{\alpha\text{-sat}}$ nucleosome (classes 1 and 2), CENP-A$^{\alpha\text{-sat}}$ nucleosome (classes 1, 3, 5) and CENP-A$^{\alpha\text{-sat}}$ nucleosome in complex with CENP-B (classes 1, 2, 3) (see Supplementary Fig. 8). The distance between SHL7 and −7 of wrapped nucleosomes is indicated by a dashed red line. Classes showing ordered SHL 7 and SHL −7 were included in the percentage estimation for each population (class 1 for H3$^{\alpha\text{-sat}}$ nucleosome, Class 1, 2, 3 for CENP-A$^{\alpha\text{-sat}}$ nucleosomes and class 1 for CENP-A$^{\alpha\text{-sat}}$ nucleosome in complex with CENP-B. Bottom row: Classes 1, 2, 3 of CENP-A$^{\alpha\text{-sat}}$ nucleosome in complex with CENP-B, showing different levels of DNA unwrapping. For all cryoEM data collection and map refinement statistics, see Supplementary Tables 6–8. For detailed statistical information on histograms in **c**, **d** (number of traces, replicates), see Supplementary Table 9. Source data are provided as a Source Data file.

stabilize next-neighbor nucleosome interactions within chromatin structure (Fig. 6a). We reconstituted H3 or CENP-A chromatin fibers on a DNA template either with or without B-Boxes positioned in nucleosomes N5 and N7, as well as containing fluorescent labels to detect nucleosome stacking (DA1, Fig. 6b and Supplementary Fig. 1). This arrangement of B-Boxes is similar to their occurrence in native human centromeric repeats[41]. We then proceeded to titrate CENP-B to the reconstituted chromatin fibers and detected structural changes via smFRET. H3 chromatin showed a high-FRET population ($E_{FRET} = 0.56$)

in the absence of CENP-B (Fig. 6c, left and middle, Supplementary Table 9). Adding CENP-B up to a concentration of 50 nM had a negligible effect on H3 chromatin fibers lacking B-Boxes (Fig. 6c, d). However, when exposing H3 chromatin containing two B-Boxes in next-neighbor nucleosomes to increasing CENP-B concentrations, we observed a decrease in the high-FRET population and a simultaneous increase in the low-FRET population (Fig. 6c, e). Transient CENP-B binding thus results in a substantial loss of nucleosome stacking. When performing these experiments with CENP-A chromatin fibers, which

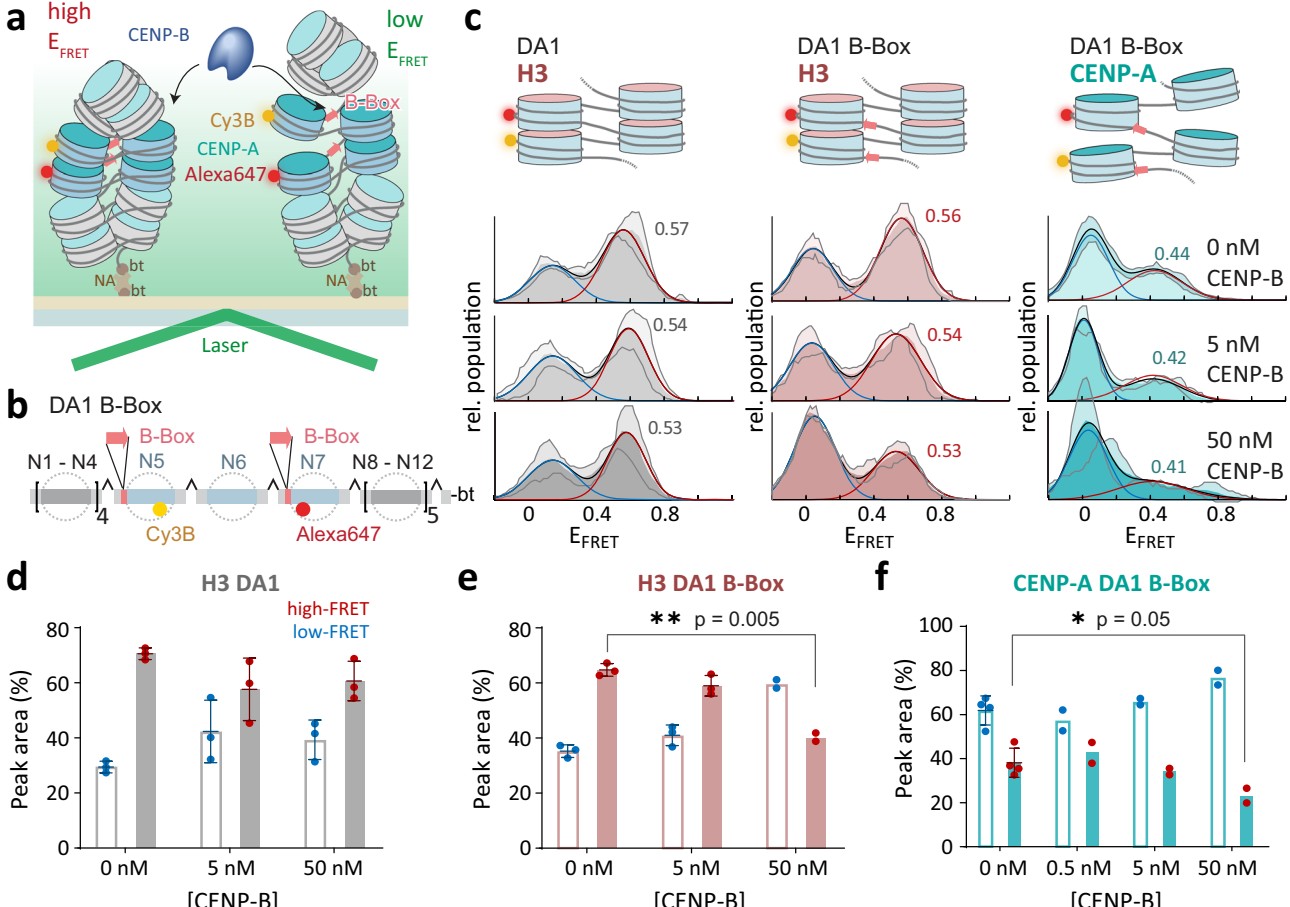

**Fig. 6 | CENP-B can induce local unstacking of chromatin. a** Scheme of smFRET experiment. **b** Scheme of chromatin DNA containing B-Box sites at nucleosome N5 and N7 with FRET donor (Cy3B, yellow) and acceptor (Alexa Fluor 647, red) dyes at nucleosome N5 and N7 (DA1BB, top). **c** Illustrative structures (left) and histograms of $E_{FRET}$ of chromatin fibers at 4 mM $Mg^{2+}$ concentration (right). Rows indicate histograms for H3 DA1, H3 DA1BB and CENP-A DA3BB, respectively. Columns indicate CENP-B concentration. Histograms are the average of $n = 3$ (DA1 H3, - B-Box), $n = 3$ (DA1 H3, + B-Box; 0–5 nM CENP-B), $n = 2$ (DA1 H3, + B-Box; 50 nM CENP-B), $n = 4$ (DA1 CENP-A, + B-Box; 0 nM CENP-B), $n = 2$ (DA1 CENP-A, + B-Box; 5–50 nM CENP-B) independent repeats. Histograms are fitted with Gaussian functions (red). Shaded region shows S.D. **d** Percentage of low- (blue) and high-FRET (red) sub-populations for H3-containing chromatin arrays lacking B-Boxes (DA1) at the indicated CENP-B concentrations. Error bars show S.D. **e** Percentage of low- (blue) and high-FRET (red) sub-populations for H3-containing chromatin arrays with B-Boxes (DA1BB) at the indicated CENP-B concentrations. Error bars show S.D. **f** Percentage of low- (blue) and high-FRET (red) sub-populations for CENP-A-containing chromatin arrays with B-Boxes (DA1BB) at the indicated CENP-B concentrations. **c–f** See Supplementary Tables 4 and 9 for detailed statistical information. For number of repeats, see **c**). Error bars show mean values +/− S.D. *$P< 0.05$, **$P< 0.01$,***$P< 0.001$ using 2-tailed unpaired $t$-test. Source data are provided as a Source Data file.

already exist in a dynamic and open state, we also observed a further depletion of the high-FRET population, resulting in an almost complete loss of detectable FRET at 50 nM CENP-B (Fig. 6c, f). Together, these effects may be due to steric restrictions arising from the large CENP-B protein invading chromatin structure[45], as well as an increase in flexibility and open structure of B-Box containing nucleosomes (Fig. 5 and S7).

In conclusion, we did not observe the stabilization of nucleosome interactions by chromatin-bound CENP-B. In contrast, our measurements indicate that CENP-B induces a more open chromatin state, which is characterized by larger inter-nucleosome distances. This may involve the formation of local loop-like structures or of alternative nucleosome interactions leading to larger inter-dye distances, or may just be a consequence of steric clashes of bound CENP-B with neighboring nucleosomes, as observed for transcription factors[45].

### CENP-A stabilizes CENP-B binding in cells

Taken together, our results suggest CENP-B recruitment to the centromere is affected by the underlying chromatin state, with the more open CENP-A-containing chromatin more easily able to recruit CENP-B. Simultaneously, CENP-B, when recruited to the B-Box, is able to induce

local opening of chromatin and stabilize alternative chromatin structures. Based on these results, we wondered whether CENP-B-chromatin interactions are stabilized by CENP-A in living cells.

We thus determined CENP-B dynamics in the presence and absence of CENP-A via fluorescence recovery after photobleaching (FRAP). The mCherry fluorescent protein was fused to CENP-B, placed under a doxycycline-inducible promoter and stably integrated into DLD-1 human adenocarcinoma cells (Supplementary Fig. 9a). Upon induction with doxycycline, CENP-B was expressed and accumulated at centromeres (Supplementary Fig. 9b). Careful calibration of doxycycline induction conditions allowed us to express mCherry-CENP-B at near native levels (Supplementary Fig. 9c). Importantly, the engineered DLD-1 cells further carried an AID-tag fused to endogenous CENP-A[74], allowing rapid degradation of the centromeric histone upon addition of an auxin inducer (indole-3-acetic acid, IAA), as verified via fluorescence imaging and immunofluorescence (IF) analysis (Supplementary Fig. 10a). CENP-A degradation resulted in a slight decrease in the centromeric localization of CENP-B (Supplementary Fig. 10b). Moreover, to minimize cell cycle-dependent heterogeneity of CENP-B dynamics[75] we arrested cells in the G1 phase of the cell cycle using a double-thymidine block (Supplementary Fig. 10c). Having thus established an

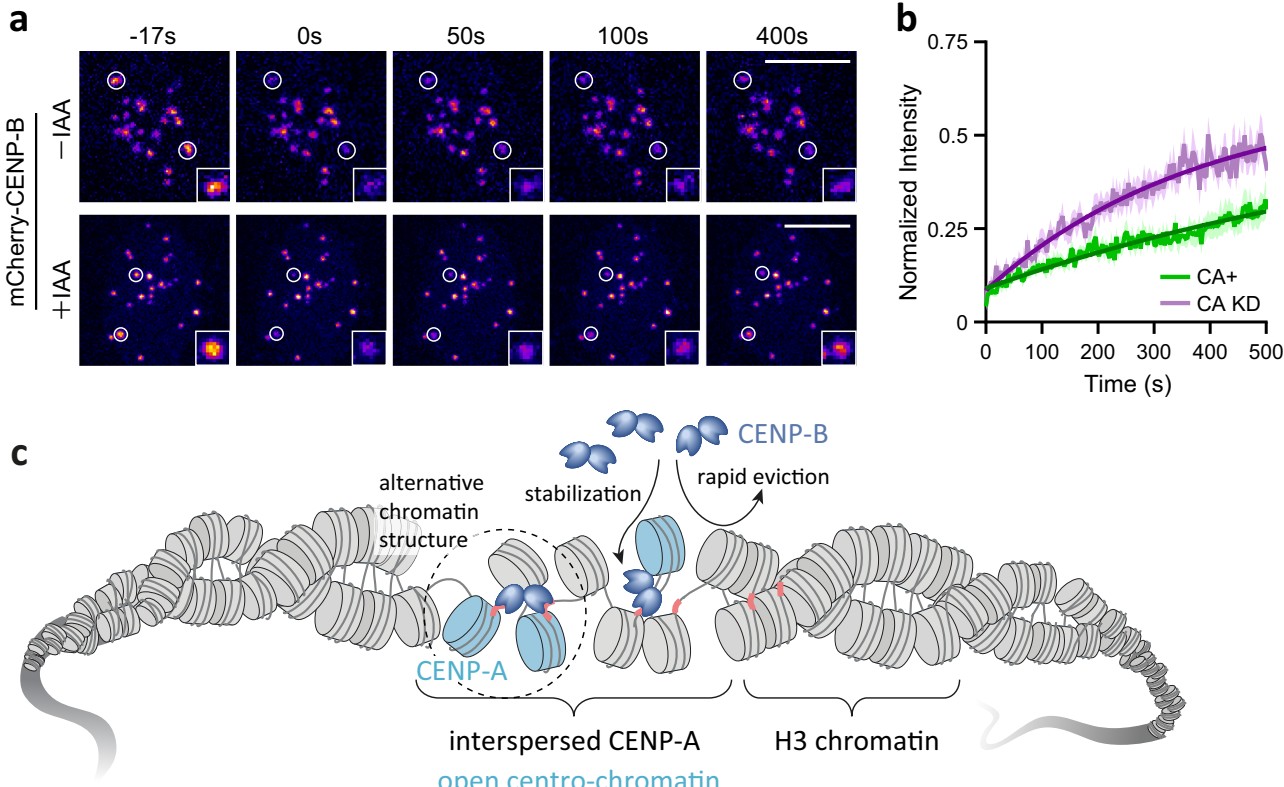

**Fig. 7 | FRAP analysis of CENP-B in the presence and absence of CENP-A depletion in G1 cells. a** Representative FRAP analysis of mCherry-CENP-B in the presence (top, -IAA) or absence (bottom, 4 h after IAA treatment) of CENP-A. Bleached areas are indicated with white circles. T < 0 frame is before bleaching and signal recovery is shown at indicated time points. Scale bar: 10 μm **b** Normalized recovery curves for quantitative FRAP measurements of CENP-B with (CA+) or without CENP-A (CA KD). Thick lines indicate mean values and the colored areas indicate the standard error. For each condition, 10 cells and 20 centromeres from two independent experiments were used for analysis. The darker lines are exponential fits to the data, with lifetimes $\tau_{CA+} = 971 \pm 225$ s, $\tau_{CA\_KD} = 381 \pm 28$ s. **c** Model of the collaboration between CENP-A and CENP-B to generate an open centro-chromatin state. Source data are provided as a Source Data file.

experimental workflow (Supplementary Fig. 10d), we determined of CENP-B dynamics in G1 via FRAP in the presence or absence of CENP-A. Here, we bleached two kinetochores per cell over 10 cells for each condition, carefully selecting cells with near-native CENP-B expression levels (Fig. 7a). This is important, as the expression level had a distinct influence on observed CENP-B dynamics: FRAP experiments in highly-expressing cells revealed highly augmented CENP-B recovery kinetics compared to CENP-B at native levels (Supplementary Fig. 10e). Analyzing average time traces of CENP-B recovery in the presence of CENP-A revealed slow-exchange with a time constant, $\tau_{CA+} = 971 \pm 225$ s, and an immobile fraction of $43 \pm 1$ % (Fig. 7b). This is comparable to earlier reports of CENP-B that showed a slow-exchange process with a time constant of 17 min in G1[75]. We then depleted CENP-A via addition of IAA for 4 h and performed FRAP of fluorescently labeled CENP-B in these CENP-A-depleted cells (Fig. 7a). Indeed, we observed a faster recovery with $\tau_{CA\_KD} = 381 \pm 28$ s and an immobile protein fraction of $42 \pm 2$ %, indicating a weakening of CENP-B binding to chromatin (Fig. 7b). Together, our results thus show that CENP-B chromatin interaction dynamics depend on the presence of CENP-A, creating a permissive chromatin environment (Fig. 7c).

## Discussion

CENP-A-containing nucleosomes lie at the core of centromeric chromatin organization, as they form an interaction hub for the CCAN and kinetochore complexes[13,76,77]. Several centromere proteins, in particular CENP-C, CENP-N and CENP-B, have been reported to directly contact CENP-A nucleosomes, and thereby may act as anchors for the CCAN on chromatin[8,14]. These interactions may lead to a distinct chromatin organization that allows complex multivalent interactions required for kinetochore formation.

Here, we investigated the dynamic organization of centromeric chromatin using single-molecule fluorescence approaches and cryoEM. Measuring FRET between neighboring nucleosomes, we observed that CENP-A renders chromatin structure open and highly dynamic. Next-neighbor stacking interactions between nucleosomes were reduced and highly transient, revealing large-scale fluctuations on the seconds timescale, compared to the more rapid, local and small-amplitude conformational fluctuations within H3 chromatin[44]. However, our FRET measurements also revealed that the linker DNA extending from CENP-A nucleosomes was ordered within a fiber context, in contrast to the greatly flexible DNA linkers observed in single nucleosomes[21,22]. Together, these results show that CENP-A containing chromatin can adopt a regular higher-order structure, but due to the dynamics of CENP-A nucleosomes, nucleosome stacking interactions are transient and fiber access is greatly increased. This effect is most likely further pronounced in native centromeres, as alpha-satellite DNA found in human centromeric repeats is less strongly nucleosome bound compared to engineered 601 DNA. Moreover, human centromeric DNA can form alternative hairpin structures, which most likely further distort chromatin structure[43]. Of note, we have however only considered chromatin fully occupied with CENP-A nucleosomes. In cells, centro-chromatin also contains blocks of H3 nucleosomes[58,78], resulting in different local dynamic properties along the centromere.

Centromere proteins that require direct access to CENP-A nucleosomes or centromeric DNA, such as CENP-B, compete against nucleosome-nucleosome or nucleosome-DNA contacts within a

chromatin fiber. This affects their binding properties, e.g., by limiting their residence times. CENP-A chromatin with its increased dynamics releases this inhibition and facilitates access to nucleosomes and linker DNA, as shown here for CENP-B. In addition, direct interactions between CENP-B and CENP-A have been proposed[38]. However, in our cryoEM studies we could not detect direct stabilization of CENP-B binding via CENP-A-mediated interactions in a mononucleosome context. Conversely, we found that CENP-B itself further opens chromatin structure upon binding. Our FRET and cryoEM analyses showed that CENP-B directly unwraps DNA from the nucleosome surface. Importantly, a comparative cryoEM analysis allowed us to determine the impact of DNA sequence on nucleosome conformation and CENP-B dependent chromatin opening. This revealed that nucleosomes containing either 601 or alpha-satellite DNA behave similarly, but effects of CENP-B were more pronounced in an alpha-satellite sequence environment, which is generating more dynamic nucleosomes.

Beyond single nucleosomes, our smFRET measurements showed that CENP-B incorporation into chromatin fibers destabilized inter-nucleosome-stacking interactions, resulting in a more open and flexible fiber conformation. B-Boxes occur frequently, with a regular spacing, in natural alpha-satellite repeats[41], allowing dimeric CENP-B to form chromatin loops[42,43]. An increased conformational flexibility of centro-chromatin, as observed in our experiments, can support such multivalent interactions and thereby promote alternative chromatin conformations.

Taken together, CENP-A and CENP-B thus collaborate to establish a more accessible centromeric chromatin state, in particular on centromeric DNA repeats. Nevertheless, CENP-B binding is only hindered but not excluded from H3 chromatin. In agreement with these observations, in cells, CENP-B is found throughout the centromere and is not limited only to CENP-A nucleosomes[28,79]. Still, when depleting CENP-A from cellular chromatin, we observed an increase in the dynamics of CENP-B, indicating weaker chromatin binding. This is in agreement with earlier studies that showed a loss of CENP-B signal at centromeres in CENP-A-depleted cells[79], underlining the fact that CENP-B retention at the centromere is reduced by CENP-A loss. Of note, the observed recovery rate constants of ~16 min in the presence and ~6 min in the absence of CENP-A are much longer than what we observed using defined chromatin in vitro. This indicates that further interactions stabilize CENP-B at centromeres in cells, compared to a reconstituted model system.

The increased flexibility of CENP-A-containing chromatin, in particular on alpha-satellite repeat DNA, is likely an important factor for CCAN recruitment by allowing biochemical access to both the CENP-A nucleosome as well as the linker DNA. This is important because recent cryo-EM studies have shown that the structured CCAN complex requires access to linker DNA which is incompatible with tightly compacted chromatin[19,20]. In contrast, it has been reported that CCAN member CENP-N (in the absence of other CCANs) induces the compaction of CENP-A-containing arrays[18] by bridging and thereby stabilizing nucleosome interactions, and overcoming the inherent flexibility of CENP-A chromatin. Thus, it is still an open question how chromatin is organized at centromeres and whether chromatin structure is dynamically coupled to centromere function during the cell cycle. Interestingly, while CENP-N stabilizes interphase recruitment of key kinetochore proteins such as CENP-C[8,80], its levels are markedly reduced at the centromere during kinetochore assembly in mitosis, correlated with structural changes in centro-chromatin[81,82].

Together, our studies show how histone variants such as CENP-A can reorganize chromatin structure, and thus promote chromatin invasion by DNA binding proteins, including CENP-B. This sets the stage for large-scale organization of centromeric chromatin, including the formation of long-range loops stabilized by CENP-B dimerization. The elastic and dynamic chromatin state, established by CENP-A and

CENP-B, then allows the binding of CCAN members, attachment of the kinetochore and buffering of the forces exerted upon chromosomes during cell division.

## Methods

### In-vitro biochemical experiments

**Expression and purification of JF549-CENP-B.** The ybbR-CENP-B-His$_6$, and ybbR-CENP-B$_{1-150}$-His$_6$ constructs were cloned into pET-15b and expressed in Rosetta-Gami 2 E. coli cells. Typically, 1 L cultures of bacteria was grown to OD = 0.8 (600 nm) at 37 °C before induction with 50 µg/mL IPTG and subsequently left overnight at 16 °C with shaking. Bacteria were then harvested by centrifugation ($3000 \times g$, 5 min), resuspended in 10 mL/L culture of resuspension buffer (phosphate buffered saline (PBS, 137 mM NaCl, 2.7 mM KCl, 8 mM Na$_2$HPO$_4$, and 2 mM KH$_2$PO$_4$) supplemented with 500 mM NaCl, and protease inhibitors (Roche)), and lysed by sonication. Post-sonication cell debris was removed by centrifugation ($100,000 \times g$, 45 min) and the resulting supernatant was filtered through a 0.2 µm filter.

The cleared cell lysate was passed through a 5 mL Histrap column (GE, AKTA system) pre-equilibrated in resuspension buffer. The His-column was washed in wash buffer 1 (10 mM Tris-HCl (pH 7.5), 2 M NaCl, 1 mM DTT) to remove any DNA contamination and wash buffer 2 (10 mM Tris-HCl (pH 7.5), 500 mM NaCl, 1 mM DTT, and 20 mM Imidazole) to remove non-specifically bound proteins until a stable UV signal was seen on the FPLC system. The column was then eluted using a 10 CV (column volume) gradient elution from wash buffer 2 to elution buffer (10 mM Tris-HCl (pH 7.5), 500 mM NaCl, 1 mM DTT, and 500 mM Imidazole). Fractions containing CENP-B were identified by SDS-PAGE and the presence of CENP-B protein was confirmed by immune-blotting. Fractions were pooled and concentrated to ~250 µL via Amicon 50 K MWCO centrifugal spin filters. The concentrated protein was then dialyzed into storage buffer (10 mM Tris-HCl (pH 7.5), 500 mM NaCl, 1 mM DTT, 10% glycerol) and the final concentration was determined by UV spectroscopy.

For fluorescent labeling, CENP-B was diluted to a final concentration of 5 µM in reaction buffer (50 mM HEPES (pH 7.5), 10 mM MgCl$_2$, 1 µM SFP synthase, 10 µM CoA-JF549) and incubate overnight at 4 °C. The reaction mixture was centrifuged at 13,000 rpm, 5 min using a tabletop centrifuge (Eppendorf) to remove aggregates before performing size exclusion purification on a Superdex200 10/300 or Superdex75 10/300 column (GE) for CENP-B or CENP-B$_{1-150}$ respectively.

Fractions were analyzed using SDS PAGE. Clean samples were pooled, concentrated, and flash frozen for storage at −80 °C. Labeling efficiency and concentration were measured by UV spectroscopy. The labeling efficiency with JF549 dye was approx. 50%, as judged from UV-VIS spectroscopy (Supplementary Fig. 3).

### Expression and purification of recombinant histones

CENP-A tetramers were purified from E. Coli as described[18,83], or purchased from EpiCypher (Cat. No. 16-0010). Canonical histones were expressed and purified as described before[59]. Briefly, individual wild-type human histones were cloned into pET-15b plasmid vectors and expressed in BL21 DE3 plysS cells. Cells were grown in LB media containing 100 µg/mL ampicillin and 35 µg/mL chloramphenicol at 37 °C until OD600 = 0.6. Expression was induced by IPTG addition to a final concentration of 0.5 mM. After 3 h expression, cells were harvested by centrifugation and resuspended in lysis buffer (20 mM Tris pH 7.5, 1 mM EDTA, 200 mM NaCl, 1 mM βMe, Roche protease inhibitor) and frozen. Cells were lysed by freeze-thawing and sonication. Inclusion bodies were harvested by centrifugation. The inclusion body pellet was washed once with 7.5 mL of lysis buffer containing 1% Triton and once without. Inclusion body pellets were resolubilized in resolubilization buffer (6 M GdmCl, 20 mM Tris pH 7.5, 1 mM EDTA, 1 mM β-mercaptoethanol (βMe)) and dialyzed into urea buffer (7 M urea,

10 mM Tris, 1 mM EDTA, 0.1 M NaCl, 5 mM 1 mM βMe, pH 7.5). Histones were purified by cation exchange chromatography using a HiTrap SP HP 5 mL column (GE Healthcare). Fractions were analyzed by SDS-PAGE and pooled, followed by dialysis into water and lyophilization. Final purification was performed by preparative RP-HPLC. Purified histones were lyophilized and stored at −20 °C until used for octamer refolding.

## Dimer and octamer refolding

In a typical octamer refolding reaction, 0.4 mg of each of the pure lyophilized human histones were dissolved in unfolding buffer (6 M GdmCl, 10 mM Tris, 5 mM DTT, pH 7.5) to an expected concentration of 2 mg/mL. The exact concentration was determined by UV spectroscopy, using the following extinction coefficients: $\varepsilon_{280nm,H2A} = 4470\,M^{-1}cm^{-1}$, $\varepsilon_{280nm,H2B} = 7450\,M^{-1}cm^{-1}$, $\varepsilon_{280nm,H3} = 4470\,M^{-1}cm^{-1}$, $\varepsilon_{280nm,H4} = 5960\,M^{-1}cm^{-1}$. For dimer refolding, equimolar amounts of H2A and H2B were mixed and unfolding buffer was added to a final histone concentration of 1 mg/mL. For octamer refolding, equimolar amounts of H3 and H4 were mixed along with 1.05 equivalents of H2A and H2B and unfolding buffer was added to a final histone concentration of 1 mg/mL. Dimers or octamers were then refolded by dialysis against refolding buffer (2 M NaCl, 10 mM Tris, 1 mM EDTA, 5 mM DTT, pH 7.5). The refolded dimers or octamers were subsequently purified by gel filtration on a Superdex S200 10/300GL column. Collected fractions were analyzed by SDS-PAGE, and octamer containing fractions were pooled and concentrated. Finally, glycerol was added to a final concentration of 50%, concentrations were determined by UV spectroscopy and octamer stocks were stored at −20 °C.

## Oligonucleotide labeling

Fluorescently labeled oligonucleotides were generated as described before[44]. Briefly, 5–10 nmol of single-stranded oligonucleotide, containing amino modifier C6 dT (purchased from Integrated DNA Technologies IDT), was diluted in 25 µl labeling buffer (0.1 M sodium tetraborate, pH 8.5). 5 µl of a 5 mM stock of succinimidyl-ester modified fluorophore (Alexa 647 or Cy3B) were added to the reaction mix and left shaking at 4 °C overnight. For a table enumerating all labeled oligonucleotides see Supplementary Table 2. The reaction progress was monitored by RP-HPLC using a gradient from solvent A (95% 0.1 M triethylammonium acetate (TEAA) pH 7, 5% acetonitrile) to solvent B (70% 0.1 M TEAA pH 7, 30% acetonitrile) on a 3 µm 4.6 × 150 mm InertSustain C18 column (GL sciences) over 20 min. More dye was added when required. For purification, the labeled DNA was precipitated via the addition 0.3 M NaOAc pH 5.2 followed by 2.75 equivalents of cold ethanol (ethanol precipitation), followed by centrifugation at $20,000 \times g$ at 4 °C for 20 min. This was repeated twice to remove excess unconjugated dye. The DNA pellet was finally dissolved in 100 µl solvent A and purified by HPLC. The purified DNA was finally isolated by ethanol precipitation and dissolved in milliQ water to a concentration of ~10 µM.

## Preparation of mononucleosomes

Labeled nucleosome DNA was prepared by PCR (fragment 1 × 601 and 1 × 601BB. Sequences in Supplementary Table 2). Nucleosomes (H3 containing 1 × 601, 1 × 601BB, and CENP-A containing 1 × 601, 1 × 601BB) were prepared following ref. 84. Typically, 1–5 µg of labeled and biotinylated DNA (1 × 601, 1 × 601BB) was combined with purified refolded H3 containing histone octamers or a mixture of 2 eq. H2A/H2B dimers and 0.9 CENP-A/H4 tetramers at experimentally determined ratios (1:1 to 1:3, DNA to histone octamer ratio) in 20 µl TE (10 mM Tris-HCl pH 7.5, 1 mM EDTA) supplemented with 2 M KCl. In a typical dialysis, the mixture was added to a micro-dialysis unit (Thermo Scientific, Slide-A-Lyzer – 10,000 MWCO), then dialyzed into TE buffer (10 mM Tris pH 7.5, 0.1 mM EDTA pH 8.0) with a linear gradient from

2 M to 10 mM KCl for 16–18 h, and finally kept in TEK10 buffer (10 mM Tris pH 7.5, 0.1 mM EDTA pH 8.0, 10 mM KCl) for another 1 h. Samples were then centrifuged at $20,000 \times g$ for 10 min at 4 °C and the supernatant was kept on ice. To determine the quality of MN assemblies, 5% polyacrylamide gels were analyzed using native PAGE in 0.25 x TB at 90 V on ice for 90 min (Supplementary Fig. 3a, b).

## Electrophoretic mobility shift assays (EMSA)

EMSAs to determine CENP-B binding to DNA or mononucleosomes were performed in single-molecule imaging buffer (10 mM Tris pH 7.5, 130 mM KCl, 3.2% w/v glucose), with 20 µL total volume. Typically, 200 nmol stocks of DNA and 3 µM stocks of CENP-B were prepared and serially diluted to desired concentrations. Reactions were mixed by pipetting and incubated for 30 min at room temperature. Sucrose was added to a final concentration of 8% and reactions were loaded onto 5% polyacrylamide gels run in 0.25 x TBE at 90 V for 90 min. Images were taken using ChemiDoc MP (Biorad) (Supplementary Fig. 5).

## Plug and play synthesis of labeled 12 × 601 DNA

Singly-labeled and biotinylated 12 × 601 DNA was produced as shown in Supplementary Fig. 1a. In short, recombinant 12 × 601 arrays were generated containing sites for nicking endonucleases Nb.BssSI and Nt.BstNBI (NEB) (sequences in Supplementary Table 1). Typically, 100 µg of 12 × 601 containing plasmids was digested with EcoRV, HindIII-HF, and Calf alkaline phosphatase in Cutsmart buffer (NEB) to release the 12 × 601 DNA. The DNA fragments were then purified by PEG precipitation, followed by PEG removal using Qiaquick PCR purification spin columns (Qiagen) to generate ~50 µg of array DNA. The arrays were subsequently treated with Nb.BssSI and Nt.BstNBI (NEB) at 50 °C for 2 h to generate nicked sites. 10-fold excess of complimentary oligonucleotides (Supplementary Table 2) were added to the nicked arrays and annealed by heating to 90 °C and cooling by 1 °C/min until 25 °C. Nicked sites on annealed arrays were sealed by T4 DNA ligase treatment (NEB) for 1 h in Cutsmart buffer supplemented with ATP. Biotin-containing anchors were ligated to the HindIII site. Labeling was checked by agarose gel electrophoresis (Supplementary Fig. 1a) and labeled array DNA was purified by PEG precipitation and PCR purification spin columns. Final concentration was determined by UV spectroscopy. Typical yield for 100 µg starting plasmid was ~20 µg of labeled arrays.

## Reconstitution of 12 × 601 chromatin fibers

Labeled array DNA was prepared by plug and play synthesis as described above (Sequences in Supplementary Tables 1–3). Chromatin arrays (H3 containing chromatin on 12 × 601 DA1, DA1BB, DA3BB DNA (BB: B-B-Box), and CENP-A containing chromatin on 12 × 601 DA1BB, 12 × 601DA3BB DNA) were prepared following ref. 84. Typically, 100–200 pM of labeled and biotinylated 12 × 601 DNA was combined with purified refolded H3 containing histone octamers or a mixture of 2 eq. H2A/H2B dimers and 0.9 CENP-A/H4 tetramers at experimentally determined ratios (1:1 to 1:3, DNA to histone octamer ratio) in 30 µl TE (10 mM Tris-HCl pH 7.5, 1 mM EDTA) supplemented with 2 M KCl. H3 containing arrays were also supplemented with 0.5–1 eq. MMTV DNA to prevent octamer over-saturation. In a typical dialysis, the mixture was added to a micro-dialysis unit (Thermo Scientific, Slide-A-Lyzer–10,000 MWCO), then dialyzed into TE buffer (10 mM Tris pH 7.5, 0.1 mM EDTA pH 8.0) with a gradient from 2 M to 10 mM KCl for 16–18 h, and finally equilibrated in TEK10 buffer (10 mM Tris pH 7.5, 0.1 mM EDTA pH 8.0, 10 mM KCl) for another 1 h. Samples were then centrifuged at $20,000 \times g$ for 10 min at 4 °C and the supernatant was kept on ice. To determine the quality of array assemblies, they were analyzed using native PAGE on 0.6% Agarose 0.25 x TB gels, run at 90 V on ice for 90 min (Supplementary Fig. 1b, c). Chromatin assembly quality was further verified by ScaI digestion of 12x assemblies. Only samples

showing full saturation and minimal free 601 DNA were used for further experiments.

## Preparation of microfluidic chambers for smFRET/TIRF experiments

Coverslips and glass slides were cleaned, silanized and PEGylated[59]. Briefly, coverslips (24 × 40 mm, 1.5 mm thickness) and glass slides (76 × 26 mm with 2 rows of 4 holes drilled) were sonicated for 20 min in 10% Alconox, rinsed with milliQ water and the procedure was repeated sequentially with acetone and ethanol. Both coverslips and glass slides were then placed in piranha etching solution (25% v/v 30% $H_2O_2$ and 75% v/v $H_2SO_4$) for a minimum of 2 h. After thorough washing with milliQ $H_2O$, coverslips and slides were sonicated in acetone for 10 min, then incubated with 2% v/v aminopropyltriethylsilane (APTES) in acetone for 15 min, and dried. Flow-chambers were assembled from one glass slide and one coverslip separated by machine-cut double-sided 0.12 mm tape (Grace Bio-labs) with channels between each opposing hole in the glass slide. Pipette tips were fitted in each of the 2 × 8 holes on each side of the silanized glass flow chambers as inlet reservoir and outlet sources and glued in place with epoxy glue. The glue was allowed to solidify for 30–40 min. Subsequently, 350 µL of 0.1 M tetraborate buffer at pH 8.5 was used to dissolve ~1 mg of biotin-mPEG(5000 kDa)-SVA (Layson Bio, Cat. No. Biotin-PEG-SVA-5000–100 mg), and 175 µL from this was transferred to 20 mg mPEG (5000 kDa)-SVA. This was centrifuged and mixed to homogeneity with a pipette before 10–20 µL aliquots were loaded into each of the eight channels in the flow chamber. The PEGylation reaction was allowed to continue for the next 2½–4 h after which the solution was washed out with degassed ultra-pure water (Romil).

## Single-molecule TIRF co-localization microscopy measurements

Measurements were performed according to established protocols[59]: Objective-type smTIRF was performed using a fully automated Nikon Ti-E inverted fluorescence microscope, equipped with an ANDOR iXon EMCCD camera and a TIRF illuminator arm, controlled by NIS-elements and equipped with a CFI Apo TIRF 100 x oil immersion objective (NA 1.49), resulting in a pixel size corresponding to 160 × 160 nm. Laser excitation was realized using a Coherent OBIS 640LX laser (640 nm, 40 mW) and Coherent OBIS 532LS laser (532 nm, 50 mW) on a custom-made laser bench. Wavelength selection and power modulation was done using an acousto-optical tunable filter (AOTF) controlled by NIS-elements. Typical laser intensities in the objective used for measurements were 0.8 mW for both 532 nm and 640 nm laser lines. For all smTIRF experiments, flow channels were washed with 100 µL degassed ultrapure water (Romil), followed by 100 µL 1 x T50 (10 mM Tris pH 8, 50 mM NaCl). Channel quality was checked by observing background fluorescence using both 532 nm and 640 nm excitation. 50 µL of 0.2 mg/mL neutravidin was then injected and incubated for 5 min, followed by washes using 200 µL of 1 x T50 buffer. 50 pM of Alexa647 labeled DNA/mononucleosomes/12-mer chromatin assemblies were then flowed in for immobilization in T50 with 2 mg/mL bovine serum albumin (BSA, Carl Roth). A 25 × 50 µm imaging area was monitored using 640 nm excitation to check for sufficient coverage. 200 µL 1 x T50 was used to wash out unbound Alexa647 labeled DNA/mononucleosomes/12-mer chromatin assemblies. 5–10 nM JF-549 labeled CENP-B or CENP-B$_{1-150}$ (see table below for details) was flowed in using imaging buffer (50 mM Tris pH 7.5, 40 mM KCl, 110 mM NaCl 10% v/v glycerol, 0.005% v/v Tween 20, 2 mM Trolox, 2 mM nitrobenzyl alcohol (NBA), 2 mM cyclooctate-traene (COT), 3.2% w/v glucose, 1x glucose oxidase/catalase oxygen scavenging system and 2 mg/mL BSA). Images were recorded using the following parameters where $t_{on}$ denotes the camera integration time, and $t_{off}$ indicates interspersed time intervals of camera inactivity: $t_{on}$ = 100 ms, $t_{off}$ = 0.3 ms, Imaging cycle: orange channel: 99 frames, far-red channel: 1 frame, n repeats: 40.

## Colocalization data analysis

Single-molecule trace extraction and trace analysis were using an established pipeline[45,59] with some adjustments. Acquired movies were background-corrected in ImageJ using a rolling ball algorithm. Trace extraction and analysis was performed in custom-written MATLAB software: DNA/nucleosome or chromatin positions were detected via a local maxima approach. Sequential images were aligned using the far-red channel to compensate for stage drift. Fluorescence intensities (in the orange channel) were extracted from the stack within a 2-pixel radius of the identified DNA peaks. Every detected spot in the orange channel was fitted with a 2D-Gaussian function to determine co-localization with immobilized DNA/chromatin. Peaks exceeding an experimentally determined PSF width for a single JF-549 molecule were excluded from further analysis. For noise reduction, extracted fluorescence traces were filtered using a forward-backward non-linear filter[85].

Individual binding events were detected using a thresholding algorithm. Overlapping multiple binding events were excluded from the analysis. For each movie, cumulative histograms were constructed from detected bright times ($t_{bright}$) corresponding to bound CENP-B molecules. The cumulative histograms from traces corresponding to individual DNA / mononucleosome / chromatin fibers were fitted with di-exponential functions:

$$y = \sum_{i=1}^{2} A_i \exp\left(-t/\tau_{off,i}\right) \tag{1}$$

yielding non-specific residence times $\tau_{off,0}$ or the specific residence times $\tau_{off,1}$ and $\tau_{off,2}$. Cumulative histograms constructed from dark times ($t_{dark}$), in between binding events, were fitted with mono-exponential functions:

$$y = A \exp\left(-t \cdot k_{on,app}\right) \tag{2}$$

to obtain apparent on-rate constants.

## smFRET measurements

Flow cell preparation and chromatin loading for smFRET was performed as described above, and using established protocols[44,54]. Experiments were performed in FRET imaging buffer (40 mM KCl, 50 mM Tris pH 7.5, 2 mM Trolox, 2 mM nitrobenzyl alcohol (NBA), 2 mM cyclooctatetraene (COT), 10% glycerol and 3.2% glucose) supplemented with GODCAT (100 x stock solution: 165 U/mL glucose oxidase, 2170 U/mL catalase). Experiments on the effect of CENP-B on chromatin compaction were supplemented with 2 mg/ml BSA to prevent non-specific interactions. smFRET data acquisition was carried out with a micro-mirror TIRF system (MadCityLabs) using Coherent Obis Laser lines at 405 nm, 488 nm, 532 nm and 640 nm, a 60x NA 1.49 Nikon CFI Apochromat TIRF objective (Nikon) as well as an iXon Ultra EMCCD camera (Andor), operated by custom-made Labview (National Instruments) software. For general smFRET imaging, a programmed sequence was employed to switch the field of view to a new area followed by adjusting the focus. Subsequently 2000 frames at 100 ms integration time were recorded under alternating excitation (ALEX) with using a 532 nm and 640 nm laser line. Each experiment ended with a strong laser pulse in the donor and acceptor channel to bleach the dyes for better background determination. Moreover, each experiment was repeated several times (see Supplementary Tables 4 and 6 for number of repeats), using at least two independently produced chromatin preparations on two different days.

## smFRET data analysis

Similar to colocalization experiments, acquired movies were background-corrected in ImageJ, whereas trace extraction and analysis was performed in custom-written MATLAB software. Single-molecule

fluorescence peaks were automatically detected in the initial acceptor image prior to donor excitation and the same peaks were selected in the donor channel. The donor and the acceptor images were aligned using a transformation matrix generated from 4–6 peaks appearing in both the donor and the acceptor channels. Peaks tightly clustered, close to the edges or above a set intensity threshold in either the donor or the acceptor channels were removed from analysis. All remaining traces were extracted for further analysis. Using the ALEX data, traces that showed less or more than exactly one donor or acceptor dye (due to clustering of molecules or incomplete labeling) were excluded from the analysis.

From traces of donor- ($F_D$) and acceptor ($F_A$) fluorescence emission intensity, FRET efficiency ($E_{FRET}$) traces are calculated as follows:

$$E_{FRET} = \frac{F_A - \beta F_D}{F_A - \beta F_D + \gamma F_D} \quad \text{where } \beta = \frac{F_{A,Donly}}{F_{D,Donly}} \quad \text{and } \gamma = \frac{\triangle F_{A,bleach}}{\triangle F_{D,bleach}} \quad (3)$$

where β denotes the bleed-through of donor fluorescence into the acceptor channel, and γ is the sensitivity ratio of the detection system for donor and acceptor fluorescence. The values of β = 0.071 and γ = 0.463 were experimentally determined for the dye pair Cy3B/Alexa647 in our experimental setup. From fluorescence time-traces, $E_{FRET}$ histograms were constructed, using a bin size of 0.02. $E_{FRET}$ histograms of each trace of length >5 s were normalized to total counts. Final histograms for each independent measurement were fitted using 2 or 3 Gaussian functions:

$$\sum_i A_i e^{-\left(\frac{x - c_i}{\sigma_i}\right)^2} \quad (4)$$

For fit values, see Supplementary Tables 4 and 6.

Traces were selected based on the following criteria: (1) Initial total fluorescence of the donor and the γ-corrected acceptor of >2000 counts over baseline (at 900 EM gain). (2) At least 5 s prior to bleaching of acceptor or donor. (3) Single bleaching event for donor or acceptor. (3.a) if acceptor bleaches first; leads to anticorrelated increase in donor to same total fluorescence level as prior to bleaching. (3.b) if donor bleaches first, the acceptor dye must still be fluorescent when directly excited. Traces that did not match these criteria were rejected from the analysis. Traces were finally sorted into "static" and "dynamic" dependent on the existence of anticorrelated intensity fluctuations in donor- and acceptor channels. Low and high-FRET populations were assigned by the Gaussian distributions obtained after fitting to the formula above (Low FRET: <0.2 $E_{FRET}$ center of the distribution, High FRET: >0.2 $E_{FRET}$ center of the distribution).

## Bulk FRET assay
1.8 μM of H3$^{α-sat}$ or CENP-A$^{α-sat}$ nucleosome was mixed with 0, 0.95 μM (0.5x) or 5.4 μM (3x) of CENP-B in 10 mM HEPES pH 7.5, 150 mM NaCl, 2 mM DTT and incubated on ice for 30 min. 30 μl of each reaction (in triplicates) were loaded into 384-wells plate, and the FRET measurements were conducted using BioTek Synergy™ Neo2 Multi-Mode Microplate Reader at 23 °C. The emission spectrum was captured from 550–700 nm at 1 nm intervals. The 520 nm excitation bandwidth was at 5 nm and the emission bandwidth at 20 nm. All reads had max gain, high lamp energy, and 50 reads per measurement.

## CryoEM experiments
**Preparation of monocucleosome:CENP-B complexes for cryoEM.** CENP-A and H3 nucleosomes were prepared as described above (see also ref. 47), and dialyzed against 10 mM HEPES pH 7.5, 150 mM NaCl, 2 mM DTT buffer. 6–10 μM of CENP-A nucleosome was mixed with 3-fold molar ratio of full-length CENP-B (in the same buffer) and incubated on ice for 30 min. Complex formation was monitored with a 5% native PAGE gel.

## Cryo-EM grids preparation, data acquisition and processing
3.5 μl of CENP-A nucleosome alone and in complex with CENP-B at concentrations between 0.7 and 1.2 mg/ml was applied to freshly glow-discharged Quantifoil R2/1 300 mesh grids. The grids were blotted for 5 s and frozen in liquid ethane using a FEI Vitrobot automatic plunge freezer. Humidity in the chamber was maintained at 100%.

Electron micrographs were acquired using the Titan Krios electron microscope (Thermo Fisher Scientific) at 300 kV (ScilifeLab cryo EM facility, Stockholm, Sweden). All datasets, 10971 movies for CENP-A$^{α-sat}$ nucleosome, 8366 movies for CENP-A$^{601,BB}$ nucleosome and 19619 movies for CENP-A$^{601\ BB}$ nucleosome in complex with CENP-B were acquired using a Gatan K3 electron detector at a nominal magnification of 105k and a pixel size of 0.8566 Å. The total electron exposure of 57.5 e$^-$/Å$^2$ was distributed over 40 frames. Data were acquired using EPU v2.24.0 (Thermo Fisher Scientific) automated data acquisition software with AFIS. The defocus range was from −0.6 to −2.2 μm with a step size of 0.2 μm.

CENP-A$^{α-sat}$ nucleosome in complex with CENP-B (21770 movies), and H3$^{α-sat}$ nucleosome alone (5101 movies) datasets were acquired in the cryo EM facility at UCEM (Umeå university, Sweden) using Falcon4i electron detector at a nominal magnification of 165k, a dose rate of 6.31 electrons per pixel per second and a pixel size of 0.704 Å. The total electron exposure of 50 e$^-$/Å$^2$ was distributed over 1206 frames. Data were acquired using EPU v3.2.0 (Thermo Fisher Scientific) automated data acquisition software with AFIS. The defocus range was from −1.2 to −2.4 μm with a step size of 0.3 μm.

Movie frames were aligned using patch motion in CryoSPARC 4.1.2[86]. CTF was estimated in a patch manner and several hundreds of particles were manually picked. The resulting useful particles were then used for automatic particle picking using template picking. The 2D class averages were generated in CryoSPARC 4.1.2. Inconsistent class averages were removed from further data analysis. Particle duplicates were removed, and the initial reference was generated using ab initio processing. The reference was filtered to 30 Å, and C1 symmetry was applied during a first homogenous refinement. Subsequent 3D classifications with target resolution limited to 7 Å were done in CryoSPARC 4.1.2 for all datasets (Supplementary Fig. 1). Volumes from different classes were used as references for following 3D heterogenous refinements. Particles from each class were split into two datasets and refined independently, and the resolution was determined using the 0.143 cut-off. For CENP-A$^{601\ BB}$ nucleosome in complex with CENP-B, the DNA linker region was subtracted from the particles showing extra CENP-B density and was locally refined[87] and 3D classified in Cryosparc V.4.1.2. After local refinement in Cryosparc, the subtracted particles were exported to Relion v3.12 for further 3D classifications without alignment, 3D auto-refine and post-processing[88]. Figures showing cryoEM maps were generated in Chimera[89] and ChimeraX[90].

## Cell experiments
**Generation of DLD-1 TIR1 CENP-A$^{EGFP-AID-CENP-A}$ TetOn mCherry-CENP-B cell line.** We modified a DLD-1 TIR1 CENP-A$^{EGFP-AID-CENP-A}$ cell line[74] using a PiggyBac transposon to express mCherry-CENP-B under a TetOn promoter (Supplementary Fig. 9a). To this end, we fused mCherry to the N-terminus of CENP-B (separated by a Gly-Ser linker). This construct was inserted into a PiggyBac TetOn vector (Addgene, #176482) (Supplementary Fig. 9a). Cells were plated in 6-well plates at a density of 5 × 10$^5$ cells /well and cultured in 10% FBS-DMEM the day before transfection. DLD-1 TIR1 CENP-AEGFP-AID-CENP-A cells were transfected with 2.5 μg of PiggyBac-TetOn-mCherry-CENP-B-neomycin and 0.8 μg of plasmid carrying PiggyBac transposase using Lipofectamine 3000 reagent (Thermofisher, L3000001). 6 h after transfection, the medium was changed to fresh 10% FBS-DMEM. At 48 h after medium change, positive cells were selected using fresh medium containing 1 mg/mL G418 (Sigma-Aldrich, 345812), and expanded.

After induction of mCherry-CENP-B expression with 100 ng/mL doxycycline, Sample acquisition and cell sorting was conducted on the BD FACSMelody™ cell sorter (BD Biosciences) using a 561-nm laser and a 613/18 nm emission filter. Cells were sorted for equal mCherry fluorescence via FACS and used for FRAP experiments.

## FRAP analysis of CENP-B in DLD-1 TIR1 CENP-A$^{EGFP\text{-}AID\text{-}CENP\text{-}A}$ TetOn mCherry-CENP-B cells

DLD-1 TIR1 CENP-AEGFP-AID-CENP-A TetOn mCherry-CENP-B cells were cultured in RPMI1640 medium (Nacarai Tequse, Kyoto, Japan; cat. No. 30264-85) supplemented with 10% fetal bovine serum (Sigma, cat. No. F7524, Lot No. BCBV4600) and penicillin/streptomycin (Gibco, cat. No. 15140122) at 37 °C and 5% CO2. The cells were arrested at G1-phase by double thymidine block: the cells were treated with 2 mM thymidine (Sigma, cat. No. D1895) for 24 h, released from the first block with 24 μM 2′-deoxycytidine (Sigma, cat. No. D3897) for 12 h, and then blocked again by the same condition as the first block. To confirm the G1-phase arrest of the cells, we labeled the cells with 10 μM EdU for 30 min and detected EdU-positive cells by using a Click-iT EdU imaging kit (Thermo Fisher, cat. No. C10340 as a S-phase cell marker). In addition, we immunostained the cells with an anti-H4K20me1 antibody (2 μg/mL, clone ID: 22G3, a kind gift from Dr. Hiroshi Kimura (Tokyo Tech University) as a G2/M-phase marker[91]. mCherry-CENP-B was induced by adding 50 ng/ml doxycycline (Tokyo Chemical Industry Co., Ltd., Tokyo, Japan; cat. No. D4116) for 12 h. The expression levels of mCherry-CENP-B were assessed by immunofluorescence. The cells expressing mCherry-CENP-B were also immunostained with an anti-CENP-B antibody (5 μg/mL, clone ID: 5e6c1, a kind gift from Dr. Hiroshi Masumoto (Kazusa DNA Research Institute)[40] to evaluate expression levels of mCherry-CENP-B, compared anti-CENP-B signals in cells without exogenous mCherry-CENP-B expression using Fiji software (v.1.54 f)[92]. We used the cells in which the total CENP-B level was less than two-fold compared with wild-type cells for FRAP. For CENP-A knockdown, 0.5 mM indole-3-acetic acid (IAA) (Sigma, cat. No. I5148) was added to the cells 4 h before the FRAP. The CENP-A knockdown was confirmed by microscopy before FRAP experiment and the cells in which the CENP-A levels were under the detection limit were used.

FRAP analysis was performed using a LSM780 confocal microscope system (Carl Zeiss, Jena, Germany) equipped with a 40 x C-Apo water-immersion objective lens (NA = 1.2), controlled by built-in Zen 2.3 SP1 software. We set and fixed image acquisition conditions in order to adjust the mCherry-CENP-B levels for FRAP as follows: 0.5 % transmission of a 561-nm laser, 121 ms/frame, 3.15 μs/pixel, average 1, 256 × 128 pixels, 40 μm pinhole corresponding to 1 airy unit, 4 x zoom: thus, we could analyze CENP-B dynamics in cells with similar expression levels of mCherry CENP-B. Five z-stack images with 0.9 μm intervals were acquired to compensate for the z-directional centromere movement. Five image frames were acquired before bleaching, and then two centromeres (each bleached region was ~2 μm in diameter) were bleached using 100% of a 561-nm laser with three iterations, followed by the capture of a further 200 image frames, using the original setting. After maximum intensity projection of the images, followed by correcting xy-directional movements by Stack-Reg, a plugin function of Fiji software, the fluorescence intensities in the bleached region were quantified using Fiji software. Photobleaching during imaging was monitored using the intensity of the unbleached centromere and normalized before drawing the recovery curve. Intensity graphs were normalized between the average pre-bleach intensity (1.0) and the first image after the bleach pulse (0.0). Results were averaged 20 FRAP curves for 10 cells. All graphs were generated using GraphPad Prism 10.0.0 (GraphPad Software).

## Statistics and reproducibility

For the exact number of traces as well as the number of independent replicates for all smFRET experiments of CENP-A and H3 chromatin fibers (Fig. 2), see Supplementary Table 4, for all CENP-B nucleosomes and chromatin binding experiments (Figs. 3–4), see Supplementary Table 5, for all CENP-B dependent chromatin decompaction experiments (Figs. 5–6), see Supplementary Table 9. Micrographs in Fig. 1f show typical smFRET data, and have been replicated according to the number of independent replicates in Supplementary Tables 4 and 9. Micrographs in Fig. 3a show typical single-molecule colocalization fluorescence data, and have been replicated according to the number of independent replicates in Supplementary Table 5.

## Reporting summary

Further information on research design is available in the Nature Portfolio Reporting Summary linked to this article.

## Data availability

The single molecule data generated in this study have been deposited in zenodo.org under the https://doi.org/10.5281/zenodo.8233663, https://doi.org/10.5281/zenodo.8233659, https://doi.org/10.5281/zenodo.8233667, https://doi.org/10.5281/zenodo.8233665, https://doi.org/10.5281/zenodo.8239003, https://doi.org/10.5281/zenodo.8250004, https://doi.org/10.5281/zenodo.8239482, https://doi.org/10.5281/zenodo.8239639, https://doi.org/10.5281/zenodo.8239637, https://doi.org/10.5281/zenodo.8239613. Source data for Figs. 1–7 and Supplementary Fig. 4, 6 and 9, are provided with the paper as a Supplementary File. CryoEM maps were deposited to EMDB (https://www.ebi.ac.uk/emdb/) under accession no. EMD- 18739, EMD- 18740, EMD-18745, EMD- 18753, EMD- 18763, EMD- 18768, EMD- 18775, EMD-18776. Source data are provided with this paper.

## Code availability

All custom-made computer code is available upon request from the corresponding authors.

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

## Acknowledgements

We thank Karolin Luger and Keda Zhou for CENP-A tetramers, and initial discussions on the project, Daniele Fachinetti for the donation of the DLD-1 cell line, Hiroshi Kimura for the anti-H4K20me1 antibody, Hiroshi Masumoto for the CENP-B antibody. We thank Maxime Mivelaz and Ruud Hovius for comments on the manuscript. B.F. thanks the Swiss National Science Foundation (SNSF grant no. 31003A_173169 and grant no. 310030_200604), the European Research Council (ERC Consolidator grant no. 724022) and EPFL for financial support. N.S. and A.A.A. are supported by NCMM and Research Council of Norway (grant numbers 187615 and 325528). The work in the Fukagawa Lab was supported by CREST of JST (JPMJCR21E6), JSPS KAKENHI Grant Numbers 17H06167, 20H05389, 21H05752, 22H00408, and 22H04692 to T.F., and JSPS KAKENHI Grant Numbers 20H05885, and 20H05891 to Y.H. Cryo-EM data were collected at Umeå and Stockholm Core Facilities for Electron Microscopy, nodes of the Cryo-EM Swedish National Facility, funded by the Knut and Alice Wallenberg, Family Erling Persson and Kempe Foundations, SciLifeLab, Stockholm University and Umeå University.

## Author contributions

H.N. and B.F. conceived the project. H.N. purified proteins, assembled chromatin and performed the single-molecule experiments. H.N. and B.F. designed, analyzed and interpreted the single-molecule experiments. N.S. and A.A.A. independently started cryoEM work on CENP-A nucleosomes in complex with CENP-B. A.A.A. purified proteins, prepared samples for cryoEM, processed cryoEM data and performed ensemble FRET experiments. A.A.A., M.H. and N.S. analyzed and interpreted cryoEM data. W.C. created cell lines for FRAP experiments. Y.H.

and T.F. performed the FRAP experiments. B.F. coordinated the project. All authors prepared figures and contributed to writing the manuscript.

## Competing interests

The authors declare no competing interests.
