## [Peer Review File · Nature Communications]

CENP-A and CENP-B collaborate to create an open centromeric chromatin stateREVIEWER COMMENTS

Reviewer #1 (Remarks to the Author):

Centromeres are regions of chromosomes essential for chromosomal segregation during both mitosis and meiosis. Centromere functionality is supported by a specialized centromeric chromatin characterized by the presence of the histone H3 variant CENP-A enriched over repetitive DNA and bound by CENP-B.

Previous structural studies showed that the DNA molecule is less tightly packed around CENP-A-containing nucleosomes compared to canonical H3 nucleosomes, suggesting an increased flexibility and accessibility of centromeric chromatin. However, these studies were mainly done on simplified models of mono or tri-nucleosomes and using techniques which only give a limited information on molecular dynamics in solution. So, the question whether the centromeric chromatin is more open and dynamic compared to canonical chromatin remains still open.

In the present study, the authors use a recently developed, single molecular fluorescence techniques to investigate the behavior of CENP-A nucleosomal arrays in solution. They propose that CENP-A nucleosomal arrays are more open and dynamic than the canonical H3 arrays, supporting the hypothesis of a more accessible centromeric chromatin. They also show that the binding of CENP-B to its cognate DNA sequence is facilitated in the context of CENP-A nucleosomes. Moreover, they conclude that the binding of CENP-B further opens CENP-A chromatin and that the presence CENP-A nucleosome further stabilizes CENP-B binding.

While this study is well performed and gives a valuable information on the dynamic behavior of CENP-A nucleosomal arrays in solution and on the role of CENP-B in this context, I am worried about the validity of their conclusions in respect to the natural context.

My main criticism is that the chromatin template that they used is not biologically relevant (see major criticism #1 below) and their conclusions can be mis-interpreted. This is already a critical point considering the nature of their experiments that open the ground to experimental interpretation. Beside these red flags, the authors did not raise any of the limitations of this study, often jumping to conclusions too easily and only minorly provided alternative interpretations. This can drive the readers to wrong conclusions.

Major criticisms

1) The authors use a nucleosome array that is unlikely to be biologically relevant. First, they assemble CENP-A on 601-repetitive DNA, not alpha-sat DNA. The referee understands that the FRET donor-acceptor pairs are on the DNA, but it is well established that the DNA sequence plays an important role in the formation and stability of the nucleosome itself (<https://doi.org/10.1093/nar/gks261>; <https://portlandpress.com/essaysbiochem/article/64/2/205/225902/CENP-A-nucleosome-a-chromatin-embedded-pedestal>). So, analyzing the behaviors of CENP-A chromatin in its natural environment could be very different from that from the 601 sequence. Maybe what the authors modelled is the behavior of ectopic CENP-A since CENP-A is not only localized at centromere. This can be still of interest, but it should be discussed. It is true that 601 favors nucleosome positioning, but it is now clear that alpha-satellite can do the same, at least in mononucleosomes (<https://www.ncbi.nlm.nih.gov/pmc/articles/PMC3052702/>) and that the centromeric DNA dictates a specific nucleosome phasing (<https://doi.org/10.1038/nsmb.2562>).

Additionally, the authors are using 12-mers of CENP-A nucleosomes that is unlikely (but not proven yet) to exist. This is not even discussed. For example, Takizawa et al., 2020 (<https://www.sciencedirect.com/science/article/pii/S0969212619303570?via=ihub>) prepared trinucleosomes (H3-CENP-A-H3) by ligation of mononucleosomes. Can this be done on this context to assemble 12-mers?

Finally, when they study how CENP-B modulates CENP-A chromatin they used only 1 or 2 B-boxes-containing DNA. While studying the behavior of the chromatin with only 1 B-box (as in figure 4) does not give much information in my opinion (considering that the authors themselves observed a contribution of CENP-B dimerization domain, figure 3), also placing only 2 B-boxes in a 12-mer could lead to incorrect conclusions.

2) Some of the relevant figures (as in figure 2 and 3) present N = 2. Despite this, the authors

applied statistic on their graphs. Is the statistical analysis performed on the average of independent experiment or on individual events?

Related to that, all graphs are shown as bar graphs, therefore it is not possible to determine single points variations and distributions. In some cases, this can be very informative as in figure 3I.

3) Figure 5: when CENP-B binds to 12-mer (H3 as well as CENP-A), FRET is reduced. The loss of FRET can be interpreted as loss of nucleosome stacking, and so a more flexible chromatin or less folded chromatin. Indeed, the authors conclude "our measurements indicate that CENP-B induces a more open chromatin state, which is characterized by larger internucleosome distances». From their drawing is indeed not surprising that CENP-B addition creates a more open chromatin as the distance between the "fluorescent" nucleosomes can indeed increase upon CENP-B binding. Especially if CENP-B dimerizes. And this can be considered as a more open chromatin. However, it does not mean that the nucleosomes are more flexible; they could be at a longer distance (lower FRET) but still more stable. It is surprising that the authors did not quantify single molecule traces showing large-scale fluctuations, as in Fig. 2. Having also other sensors in their DNA or CENP-B boxes at different position could help to decipher better the situation.

The authors also wrote: "Transient CENP-B binding thus results in a substantial loss of nucleosome stacking, even though the linker DNA orientation in H3 chromatin is not strongly altered under the same conditions (Figure S7)." The conditions are not exactly the same. In Figure 5 the DNA contains 2 B-boxes, in Figure S7 there is only one B-box. So the linker DNA orientation (DA3 pair) is not tested when 2 CENP-B molecules could bind to CENP-A or H3 12-mers.

Finally, it is unknown if the measurements that they do are on a monomeric CENP-B or a dimer. We do not know the CENP-B dimerization K_d , but at 50nM, the majority of CENP-B is likely in the monomeric form. Authors should repeat this experiment using a CENP-B dimerization mutant to see if chromatin dynamics and compaction changes compared to using a full-length CENP-B (at various concentrations, even higher than 50nM) or delta binding domain.

4) Figure 6: I am confused by FRAP experiment interpretation as not an expert. In WT cells the recovery time is 220 s with an immobile fraction of 38%. In CENP-A depleted, recovery time is 136 s and immobile protein fraction of 26%. The authors conclude that CENP-B interaction is weakened when CENP-A depleted. This makes sense considering the reduced immobile fraction that it means weaker interaction, but what about the recovery time? In CENP-A depleted, the chromatin is supposed to be more closed (accordingly figure 2), so CENP-B chromatin accessibility is supposed to be perturbed. Why the recovery time is faster? Is because CENP-B is more dynamic?

In my opinion this experiment should be repeated in condition where CENP-A is rapidly removed and not with a low kinetics RNAi where many things happen during the 48-72hr time of RNAi, including changing in chromatin accessibility. The authors seem to have an auxin inducible cell line from Facchinetti team, why not using it?

Minor points:

1. Along the whole text the term centromere and kinetochore are used indistinguishably. This is incorrect and it needs to be revised.
2. In the introduction the authors said that CENP-B is the only kinetochore (please correct to centromeric) protein that makes direct contact with DNA. This is incorrect as recently seen by the Cryo-EM structures. They can add "in a DNA sequence dependent manner".
3. Figure 1C: from the schematic is unclear if every nucleosome is spaced by a 30bp linker.
4. Figure 1: the authors discussed results on H3 chromatin in the text but there is no data about it in the figure. Same in the end of page 7 where they describe an experiment but not refer to any figure.
5. Figure S3A: the band visualized with GelRed does not correspond to the one visualized with Alexa 647 (is shifted). Why?
6. Figure S3D: is JF-549 fluorescence gel flipped compared to Coomassie?
7. Figure S3C-F: It is important to show the entire gels for HisTraps and gel filtrations and the absorbance at 260nm on the gel filtration profiles.
8. Figure S8: neither the immunofluorescence nor the immunoblot are very convincing.

Reviewer #2 (Remarks to the Author):

Centromeres are specialized chromatin regions, which ensure proper chromosome segregation. The centromere regions are defined by nucleosomes containing the centromere specific histone H3 variant, CENP-A. One of key differences between CENP-A nucleosome and canonical H3 nucleosome is flexibility of DNA ends. However, how the DNA flexibility contributes to centromeric chromatin structures is largely unclear. The authors measure dynamics of reconstituted CENP-A chromatin fibers using the single-molecular FRET system that they developed recently, and found that the CENP-A fibers are more dynamic and open than the canonical H3 fibers. The authors also demonstrated that dynamic CENP-A chromatin facilitates chromatin-binding of CENP-B, which recognizes and binds to the CENP-B binding motif (B-box) in centromeres. The CENP-B binding opens the DNA linkers of CENP-A nucleosomes and CENP-A chromatin. These *in vitro* experiments provide new insights into chromatin dynamic regulation in the CENP-A nucleosome fiber context. The authors tested their model based on the *in vitro* single-molecule FRET in the native centromeres in cells using FRAP. This is the key experiment to authors' conclusions. But my major concern is the FRAP experiments. The results from those experiments are in my opinion premature for publication. The author should consider and perform more FRAP experiments to support and strengthen their conclusions. I have several comments and suggestion on the *in vitro* experiments as minor concerns.

Major concerns:

Fig6, S8A,B. The authors used the overexpressed mEOS3.2-CENP-B for their FRAP experiments. The mEOS3.2-CENP-B expression levels are way too high compared with those of endogenous CENP-B. Overexpression could affect biochemical dynamics as well as physiology in the cells. Given the authors used the dox-inducible system, the expression levels of mEOS3.2-CENP-B can be controlled. FRAP analyses with near endogenous levels of mEOS3.2-CENP-B should be provided (or at least with possible lowest levels that give enough s/n ratio for FRAP).

The authors state their CENP-B FRAP results are consistent with a previous study (Hemmerich et al., 2008). In the study, Hemmerich et al. measured CENP-B mobility in detail: short-term and long term FRAP experiments in various cell cycle stages. The authors' FRAP results look consistent to the short-term FRAP in G1/S-phase cells in Hemmerich et al., e.g. ~20% immobile fraction. However, this immobile fraction is in fact mobile in long-term FRAP experiments as slow mobile population. Hemmerich et al. concluded that most of CENP-B in G1/S-phase cells are mobile but separated into two populations: slow-exchange and fast-exchange populations, and suggested that these different resident times were potentially caused by different DNA-binding modes: one directly binds to B-boxes with high-affinity and other binds to adjacent centromeric DNA. Considering this, to strengthen the authors' conclusion, the long-term FRAP experiments (data for the slow mobile fractions) should be important. In addition, more than 80% of CENP-B population is immobile in G2-phase cells. The authors should specify the cell cycle stages of cells measured in their FRAP assay.

In the manuscript, CENP-A was knocked-down using siRNAs and the CENP-A levels were examined by immunoblots. But even though CENP-A is knocked-down or -out, CENP-A protein still remains on centromeres for several days (see Fachinetti et al., 2013, about 50% of CENP-A remains on centromeres at 48 h after CENP-A knock-out). The authors should examine CENP-A levels on the centromeres to show how much CENP-A is depleted from "centromeres" in their experiments. Fachinetti et al. also showed CENP-B levels are unchanged at 48 h after CENP-A knock-out. Although no protein reduction does not mean no change in dynamics, the authors should also examine CENP-B levels on centromeres after CENP-A knock-down by immunofluorescence.

During cell cycle, new CENP-A proteins are deposited to centromere in early G1, but not in S-phase (Black et al., 2007). It means that CENP-A protein levels on a centromere are reduced into half after DNA replication. Hemmerich et al. showed that CENP-B is mobile in G1/S-phase and become immobile in G2-phase, indicating that CENP-B would associate to centromeric chromatin more stably when CENP-A levels at centromeres are reduced. This sounds inconsistent with the authors' conclusion. The difference could be possibly due to difference of cell cycle stages or experimental settings such as protein expression levels. I would highly recommend the authors

more FRAP experiments with better controls and setup to prove their model based on in vitro single-molecular FRET in the native context. It will improve and strengthen the manuscript.

Minor concerns:

Fig1F,H. how much of CENP-A fibers observed did show FRET? Some fibers should form the stacked-nucleosome structure as shown in the model in figures. But others maybe not. And also in Fig1H, what portion of traces did show E fret average > 0.2? This 0.2 form intermediated E fret defined in Fig2? More explanation for its criteria would be helpful.

Fig1H. although it is published, it would be nice to have the donor-acceptor cross-correlation analysis results of H3 control of the current dataset.

Fig2A left. the trace seems to have high ERET after 60 s, but no E fret is shown in the bottom graph. I might miss but it would be helpful if there are some explanations. The right graphs need y axis definition.

Fig2A. does "40 mM KCl" in the text mean "no Mg²⁺ (or 0 mM Mg²⁺)" in the figure? If so, the authors might want to use consistent terms through the manuscript.

Fig2A,B. are the high-FRET population centered at E fret values in the figures correct? I wanted to just make sure, because the values are different from those in the text.

Fig2C,F. It is unclear to me what is "the large-scale fluctuations". More information about how authors determined the Dynamic traces (as well as high and low FRET traces) is helpful.

Fig3F. in the text, short and long resident times are $t_{off,1}$ and 2, respectively. But in table S5 they are $t_{off,0}$ and 1. They would be 1 and 2 (or 0 and 1 in the text). The authors should use consistent terms through the manuscript. I found several other inconsistencies. The authors might want to check through the manuscript.

Fig3I. in the text authors say that "not significantly different compared to those for naked DNA (Figures 3I)". But no statistical analysis between naked DNA and H3 MN there. Authors should revise the text or add statistical analysis in the figure.

p5, l.2, "...this resulted in stable E fret ~ 0.5 over several tens of seconds [39]." Here, probably, the authors should cite their own results such as Fig2B (or add control H3 fiber results in Fig1 and cite it) instead of ref39. It is an important control to show the experimental condition used in the work is fine as previously reported.

p7, l.8 "In such an experiment, individual naked DNA molecules, reconstituted nucleosomes or chromatin fibers, all labeled...(Figure 3A)" Figure 3 showed results with mono-nucleosomes. It should be "In such an experiment, individual naked DNA molecules or reconstituted nucleosomes, all labeled...(Figure 3A)"

p13, l8. "siRNA against CENP-B" should be "siRNA against CENP-A"

Reviewer #3 (Remarks to the Author):

Nagpal and Fierz describe a biochemical basis for CENP-A and CENP-B effects on centromeric chromatin. Using fluorescence resonance energy transfer (FRET) based approaches, they probe the role of CENP-A, individually, and in collaboration with CENP-B on centromeric chromatin structure. Specifically, they assemble 12-mer nucleosome arrays and mono-nucleosomes with either CENP-A or the canonical H3 and measure the effect of neighbor nucleosome stacking and linker DNA proximity using strategically positioned donor and acceptor dyes. Such studies are important since

CENP-B is a factor involved in maintaining centromere stability, however, the biochemical role of CENP-B in shaping centromeric chromatin is not entirely clear. This is especially the case at the nucleosomal array level—the one at which this study is performed.

The authors report that nucleosome arrays assembled with CENP-A have a more dynamic and open structure relative to H3 nucleosome arrays. CENP-A alone can reduce nucleosome stacking within chromatin arrays and the linker DNA are more divergent, compared to H3 chromatin. CENP-B further opens the CENP-A chromatin structure, as indicated by a reduction in high FRET populations with increasing amounts of CENP-B addition. This result is something I found particularly intriguing. The binding of CENP-B to CENP-A chromatin is enhanced by the B-box sequence positioned at the entry/exit location of the nucleosome and with CENP-B dimerization. Interestingly, CENP-B bound CENP-A and H3 mono-nucleosomes similarly, but CENP-B accessed CENP-A chromatin better than H3 chromatin. Cell-based assays suggest that transient CENP-A depletion may increase the mobility of CENP-B, suggesting a specific interaction of CENP-B with CENP-A chromatin during mitosis. The effect of CENP-A on centromeric chromatin structure is of high interest, given that it is the basis for allowing kinetochore formation and, subsequently, proper chromosomal segregation. This work provides novel insight into how CENP-A can possibly accommodate CENP-B in centromeric chromatin and complements existing structural reports.

Major concerns:

1. Cell based-experiment: Fig. S8A shows massive overexpression of the CENP-B relative to the endogenous protein. It doesn't make sense to do the experiment with unnaturally high levels of CENP-B protein around, and I don't think the results are relevant to physiological understanding at centromeres, in their current form. I'm not sure gene replacement and/or inducible expression is necessary, but lower expression is necessary for this experiment to be relevant to what might be happening at natural centromeres.

2. Cell based-experiment: It makes no sense to use siRNA to knock down CENP-A. It has been known to complicate interpretations in functional experiments and requires a wind down (so with complete removal of CENP-A message, the protein is down to 25% after 2 cell divisions [i.e. at the 48 h timepoint of their experiment]). The centromere is slowly being affected during all of this and so interpretations of their data is further compromised. Inducible degradation of CENP-A is the standard for almost a decade since introduced by the Cleveland lab. It permits a clear way to test the effect of CENP-A removal within a few hours, or sooner, from the initiation of depletion. The experiments need to be performed in this condition.

Minor Concerns:

3. The introduction says that the 'RG-Loop' is important for CENP-N and CENP-C recruitment. For CENP-N this is now considered controversial with recent structures from the Barford and Musacchio labs. It would be helpful to mention the studies in support or denial of this mode of binding. Also, I am not aware of the data that this loop is important for CENP-C, so that should certainly be cited there.

4. CENP-B is referred to as a 'kinetochore' component. Unlike CENP-A, it is also found at human centromeres at an adjacent region of the chromosome that does not build a kinetochore. Also unlike CENP-A and other components of the centromere, CENP-B is dispensable for kinetochore formation. Maybe 'component of centromere/centromeric chromatin' is better?

5. When referring to "CENP-A chromatin", are these nucleosome arrays containing all CENP-A or only the middle tetramers are "CENP-A chromatin"? Please clarify in the text.

6. In figure 2A-B, the E FRET values in the text (page 5) and the figure do not match. The text says "... a high-FRET population centered at E FRET values of 0.44 for CENP-A and 0.54 for H3 chromatin (Figure 2A,B)" however the figure marks the E FRET peak at 0.42 and 0.56, for CENP-A and H3 respectively.

Reply to reviewers

Reviewer #1 (Remarks to the Author):

While this study is well performed and gives a valuable information on the dynamic behavior of CENP-A nucleosomal arrays in solution and on the role of CENP-B in this context, I am worried about the validity of their conclusions in respect to the natural context. My main criticism is that the chromatin template that they used is not biologically relevant (see major criticism #1 below) and their conclusions can be mis-interpreted. This is already a critical point considering the nature of their experiments that open the ground to experimental interpretation. Beside these red flags, the authors did not raise any of the limitations of this study, often jumping to conclusions too easily and only minorly provided alternative interpretations. This can drive the readers to wrong conclusions.

We appreciate the reviewer's criticism that helped us improve our study and we believe that we have adequately addressed the issues raised in our revised manuscript. To this point, we have extended our study to include a structural analysis of CENP-A nucleosomes on both 601 and natural alpha-satellite DNA in isolation as well as in complex with CENP-B using cryoEM. Moreover, we have improved the in-cell analysis with newly developed cell lines and appropriate controls. We have also clearly discussed and explained both the implications and limitations of our smFRET studies using nucleosome arrays in several locations in the results and discussion. In summary, our new experiments have confirmed our original conclusions.

We believe that this new structural data together with our dynamic single-molecule studies and discussions in the text now provide an extensive and balanced picture and allow the reader to draw the correct conclusions, i.e. that DNA sequence, centromeric histones and DNA binding proteins (i.e. CENP-B) generate an open and dynamic chromatin state shaping the centromere.

1) The authors use a nucleosome array that is unlikely to be biologically relevant. First, they assemble CENP-A on 601-repetitive DNA, not alpha-sat DNA. The referee understands that the FRET donor-acceptor pairs are on the DNA, but it is well established that the DNA sequence plays an important role in the formation and stability of the nucleosome itself (<https://doi.org/10.1093/nar/gks261>; <https://doi.org/10.1042/EBC20190074>). So, analyzing the behaviors of CENP-A chromatin in its natural environment could be very different from that of the 601 sequence. Maybe what the authors modelled is the behavior of ectopic CENP-A since CENP-A is not only localized at centromere. This can be still of interest, but it should be discussed. It is true that 601 favors nucleosome positioning, but it is now clear that alpha-satellite can do the same, at least in mononucleosomes (<https://doi.org/10.1016/j.sbi.2010.11.006>) and that the centromeric DNA dictates a specific nucleosome phasing (<https://doi.org/10.1038/nsmb.2562>).

For our smFRET experiments to function, the nucleosomes have to be positioned with base-pair accuracy, as small variations in nucleosome positioning result in potentially large changes in FRET efficiency. After extensive experiments, we were not able to produce chromatin assemblies of sufficient quality on natural alpha-satellite DNA in combination with CENP-A octamers to perform smFRET measurements. Thus, we resorted to using 601 DNA for these experiments. We have now further discussed this in the manuscript on page 4 of the revised manuscript:

"In contrast, constructs containing natural alpha-satellite repeats did not result in sufficiently well-defined nucleosomes for reliable smFRET experiments, in particular in combination with CENP-A octamers."

Instead, to further investigate the effect of DNA sequence we have performed cryoEM studies on CENP-A containing nucleosomes, assembled with 601 and alpha-satellite DNA, in the presence and absence of CENP-B. We confirm our previous findings that CENP-A nucleosomes are more dynamic than their H3 counterparts, and we find that binding of CENP-B further destabilizes DNA wrapping of

CENP-A nucleosomes, particularly in the context of alpha-satellite DNA. Because particle uniformity is a necessary requirement for cryoEM studies, we could not readily extend our analysis to nucleosome arrays using the same technique. However, we believe that the results of cryoEM analysis of nucleosomes on alpha-satellite DNA, together with our smFRET results in arrays, confirm our original findings that CENP-A contributes to more open centro-chromatin and that CENP-B further enhances this effect. Furthermore, our new results indicate that this effect is even stronger in an alpha-satellite background.

We have discussed that in the revised version of the manuscript, e.g. in the discussion on page 17:

“Our FRET and cryoEM analyses revealed that CENP-B directly unwraps DNA from the nucleosomes surface. This effect was greatly enhanced when using alpha-satellite DNA, which generates more dynamic nucleosomes.”

Additionally, the authors are using 12-mers of CENP-A nucleosomes that is unlikely (but not proven yet) to exist. This is not even discussed. For example, Takizawa et al., 2020 (<https://doi.org/10.1016/j.str.2019.10.016>) prepared trinucleosomes (H3-CENP-A-H3) by ligation of mononucleosomes. Can this be done on this contest to assemble 12-mers?

We would like to point out that using our smFRET system, we probe the conformation of the central tetranucleosome within a chromatin context. In an array, the observed tetranucleosome unit is isolated from ‘end-effects’ as it is connected to neighbors, whereas an isolated tri-nucleosome is expected to be much more dynamic. We agree that a full 12-mer stretch of CENP-A nucleosome might not represent a typical arrangement in the cell, but it is experimentally tractable and, for the reasons given above, more realistic than isolated nucleosomes. Moreover, we believe that the nature of flanking nucleosomes (beyond our FRET dyes), is not greatly influencing the behavior of the central tetranucleosome.

However, we agree with the reviewer that this issue should be more directly addressed in the manuscript. We have thus extended the results (p. 5):

“Of note, in human centromeres, not all centromeric repeats are occupied by CENP-A57. Here we monitor structure over three consecutive CENP-A nucleosomes. This thus represents a region of high local CENP-A content. Other regions of the centromere, containing substantial amounts of H3 nucleosomes, are expected to exhibit intermediate dynamics between CENP-A and H3 chromatin.”

and discussion (p. 16):

“Of note, we have however only considered chromatin fully occupied with CENP-A nucleosomes. In cells, centro-chromatin also contains blocks of H3 nucleosomes^{58,78}, resulting in different local dynamic properties along the centromere.”

Furthermore, we have also considered ligation of nucleosomes, as suggested by the reviewer (and we have successfully performed it ourselves before, e.g. in Müller et al. Nature Chem Biol, 2016). Nevertheless, in our experience it is impossible to obtain samples of sufficient quality using this method to perform single-molecule FRET experiments, in particular with unstable nucleosomes containing CENP-A.

Finally, when they study how CENP-B modulates CENP-A chromatin they used only 1 or 2 B-boxes-containing DNA. While studying the behavior of the chromatin with only 1 B-box (as in figure 4) does not give much information in my opinion (considering that the authors themselves observed a contribution of CENP-B dimerization domain, figure 3), also placing only 2 B-boxes in a 12-mers could lead to incorrect conclusions.

We agree with the reviewer that centromeric chromatin has multiple B-boxes and that this could lead to specific chromatin rearrangements. However, the generation and analysis of such arrangements in a controlled system in vitro is extremely complex, if possible at all. Instead, our experimental setup allowed us to study the specific effects of CENP-B within nucleosome arrays with a limited number of CENP-B boxes. For the experiments showing CENP-B binding (Figure 4) we indeed only included one CENP-B box in the central nucleosome. This allowed a direct comparison between the nucleosome and chromatin fiber environment (i.e. between Figure 3 and 4, as well as structural studies in Fig. 5). In these types of experiments, adding more B-boxes would potentially result in several overlapping binding events, which would render the data hard to interpret. Moreover, we observe a clear effect, as CENP-B binding unwraps the nucleosomal DNA.

In Figure 5, we are investigating changes in chromatin architecture based on CENP-B binding. Here, we added two B-boxes into the central tetranucleosome, which is the sole structure surveyed by our FRET system. Adding B-boxes beyond this central tetranucleosome will not result in any observable changes (as our method is blind toward these chromatin regions).

Moreover, to clarify, we have added a sentence discussing the consequence of our DNA design to page 10 of the results section.

“This DNA design allows to detect binding dynamics without confounding effects of multivalent CENP-B binding to neighboring B-Boxes.”

Finally, we also discussed the implications of regularly spaced B-Boxes on chromatin conformation again in the discussion section:

“Beyond single nucleosomes, our smFRET measurements showed that CENP-B incorporation into chromatin fibers destabilized inter-nucleosome-stacking interactions, resulting in a more open and flexible fiber conformation. B-Boxes occur frequently, with a regular spacing, in natural alpha-satellite repeats⁴¹, allowing dimeric CENP-B to form chromatin loops^{42,43}. An increased conformational flexibility of centro-chromatin, as observed in our experiments, can support such multivalent interactions and thereby promote alternative chromatin conformations.”

2) Some of the relevant figures (as in figure 2 and 3) present $N = 2$. Despite this, the authors applied statistic on their graphs. Is the statistical analysis performed on the average of independent experiment or on individual events?

We have applied the statistics on the averaged numbers, and report n and plot the individual data points, as well as calculated p -values, and have now included the actual data underlying the graphs in tables. This will allow the reader to make their judgement.

Related to that, all graphs are shown as bar graphs, therefore it is not possible to determine single points variations and distributions. In some cases, this can be very informative as in figure 3I.

We indeed show all individual datapoints in the figure superimposed over all bars. We have however now adjusted the color scheme to make the individual points more visible. Moreover, we provide the underlying data as tables.

3) Figure 5: when CENP-B binds to 12-mer (H3 as well as CENP-A), FRET is reduced. The loss of FRET can be interpreted as loss of nucleosome stacking, and so a more flexible chromatin or less folded chromatin. Indeed, the authors conclude “our measurements indicate that CENP-B induces a more open chromatin state, which is characterized by larger internucleosome distances». From their drawing is indeed not surprising that CENP-B addition creates a more open chromatin as the distance

between the “fluorescent” nucleosomes can indeed increase upon CENP-B binding. Especially if CENP-B dimerizes. And this can be considered as a more open chromatin. However, it does not mean that the nucleosomes are more flexible; they could be at a longer distance (lower FRET) but still more stable. It is surprising that the authors did not quantify single molecule traces showing large-scale fluctuations, as in Fig. 2. Having also other sensors in their DNA or CENP-B boxes at different position could help to decipher better the situation.

We thank the reviewer for the suggestion to quantify dynamic traces for the CENP-B titration experiments. We added the new data in Table S6. For CENP-A, we indeed see a reduction of dynamic traces for higher amounts of CENP-B. However, a major reason for this effect is the overall reduction in FRET, i.e. a loss of both dynamic and high-FRET traces as stated in the text.

To investigate the mechanism, we already use two dye pair positions, one reporting on entry- and exit DNA (Figure 5) and the other reporting on nucleosome stacking (Figure 6). Together with our new cryoEM investigations on 601 and alpha-satellite DNA, our results yield a consistent picture of the effect of CENP-B on chromatin structure: Binding of CENP-B results in partial detachment and unwrapping of nucleosomal DNA, which in turn opens chromatin structure.

We discussed this further by adding this sentence on p. 14:

“Together, these effects may be due to steric restrictions arising from the large CENP-B protein invading chromatin structure{Mivelaz, 2020 #5121}, as well as an increase in flexibility and open structure of B-Box containing nucleosomes (Figure 5 and S7).”

The authors also wrote: “Transient CENP-B binding thus results in a substantial loss of nucleosome stacking, even though the linker DNA orientation in H3 chromatin is not strongly altered under the same conditions (Figure S7).” The conditions are not exactly the same. In Figure 5 the DNA contains 2 B-boxes, in Figure S7 there is only one B-box. So the linker DNA orientation (DA3 pair) is not tested when 2 CENP-B molecules could bind to CENP-A or H3 12-mers.

Good point! We have now removed the sentence mentioning ‘same conditions’.

Finally, it is unknown if the measurements that they do are on a monomeric CENP-B or a dimer. We do not know the CENP-B dimerization K_d , but at 50nM, the majority of CENP-B is likely in the monomeric form. Authors should repeat this experiment using a CENP-B dimerization mutant to see if chromatin dynamics and compaction changes compared to using a full-length CENP-B (at various concentrations, even higher than 50nM) or delta binding domain.

We agree that this could be a further interesting experiment. However, we also agree with the reviewer that at 50 nM concentration most of CENP-B is monomeric. We therefore do not expect to see a significant difference when using a dimerization-impaired point mutant. In conclusion we believe that this additional control does not significantly add to the paper, especially given the very high time investment and complexity of performing these smFRET experiments.

4) Figure 6: I am confused by FRAP experiment interpretation as not an expert. In WT cells the recovery time is 220 s with an immobile fraction of 38%. In CENP-A depleted, recovery time is 136 s and immobile protein fraction of 26%. The authors conclude that CENP-B interaction is weakened when CENP-A depleted. This makes sense considering the reduced immobile fraction that it means weaker interaction, but what about the recovery time? In CENP-A depleted, the chromatin is supposed to be more closed (accordingly figure 2), so CENP-B chromatin accessibility is supposed to be perturbed. Why the recovery time is faster? Is because CENP-B is more dynamic?

In my opinion this experiment should be repeated in condition where CENP-A is rapidly removed and not with a low kinetics RNAi where many things happen during the 48-72hr time of RNAi,

including changing in chromatin accessibility. The authors seem to have an auxin inducible cell line from Facchinetti team, why not using it?

This point was raised by all reviewers so to properly address it, we have now performed FRAP experiments using cell lines derived from the DLD-1 cell line from the Fachinetti team, allowing CENP-A degradation within 4 h. These results, performed under much better controlled conditions, still show a mobilization of CENP-B upon CENP-A depletion (see also answers to reviewers 2 and 3).

1. Along the whole text the term centromere and kinetochore are used indistinguishably. This is incorrect and it needs to be revised.

We used the term 'centromere' to designate the region on the chromosome, marked by CENP-A, where the kinetochore attaches. Conversely, we used the term 'kinetochore' to designate the protein complex attached at centromeres. We went through the text and made sure that all terms are used according to this definition.

2. In the introduction the authors said that CENP-B is the only kinetochore (please correct to centromeric) protein that makes direct contact with DNA. This is incorrect as recently seen by the Cryo-EM structures. They can add "in a DNA sequence dependent manner".

We clarified that we mean 'direct, sequence-specific DNA contacts'.

3. Figure 1C: from the schematic is unclear if every nucleosome is spaced by a 30bp linker.

We added the corresponding information into the figure legend.

4. Figure 1: the authors discussed results on H3 chromatin in the text but there is no data about it in the figure. Same in the end of page 7 where they describe an experiment but not refer to any figure.

For Figure 1, we have added the data (which was published before in a previous manuscript)
For Page 7, we added the correct figure reference.

5. Figure S3A: the band visualized with GelRed does not correspond to the one visualized with Alexa 647 (is shifted). Why?

We corrected the figure (the band was shifted due to way the image was cropped), and we added the uncropped gels for reference.

6. Figure S3D: is JF-549 fluorescence gel flipped compared to Coomassie?

We corrected the figure and added the correct fluorescence scan. Uncropped gels are added for reference.

7. Figure S3C-F: It is important to show the entire gels for HisTraps and gel filtrations and the absorbance at 260nm on the gel filtration profiles.

We did not record 260 nm traces for the proteins during purification. However, the spectrum in Fig. S3E shows that we do not have significant DNA impurities. Finally, we added all uncropped gels for reference.

8. Figure S8: neither the immunofluorescence nor the immunoblot are very convincing.

Using new cell lines, all these experiments have been redone (see new Fig. 7 and Supplementary Figures S9-S10).

Reviewer #2 (Remarks to the Author):

The authors tested their model based on the in vitro single-molecule FRET in the native centromeres in cells using FRAP. This is the key experiment to authors' conclusions. But my major concern is the FRAP experiments. The results from those experiments are in my opinion premature for publication. The author should consider and perform more FRAP experiments to support and strengthen their conclusions. I have several comments and suggestion on the in vitro experiments as minor concerns.

We have performed new FRAP experiments using more controlled conditions, as discussed in the following. The new results are shown in Figures 7 and S9.

1. Fig6, S8A,B. The authors used the overexpressed mEOS3.2-CENP-B for their FRAP experiments. The mEOS3.2-CENP-B expression levels are way too high compared with those of endogenous CENP-B. Overexpression could affect biochemical dynamics as well as physiology in the cells. Given the authors used the dox-inducible system, the expression levels of mEOS3.2-CENP-B can be controlled. FRAP analyses with near endogenous levels of mEOS3.2-CENP-B should be provided (or at least with possible lowest levels that give enough s/n ratio for FRAP).

We thank the reviewer for pointing this out. We have now generated stable cell lines that enable mCherry-CENP-B expression via Dox induction. We then quantified CENP-B expression levels both via mCherry fluorescence (yields overall expression levels of the induced protein) as well as via IF using a CENP-B antibody (yields combined expression of endogenous and induced protein). Using this information (Supplementary Fig. S9C) we could for near-native expression of mCherry-CENP-B (not more than 2-fold over endogeneous). When measuring FRAP, we indeed detected expression level-dependent dynamics, as overexpression resulted in higher CENP-B mobility (Supplementary Fig. S10E). All the analysis was thus done at near-native levels.

This is further explained in this sentence on p. 15-16:

“Here, we bleached two kinetochores per cell over 10 cells for each condition, carefully selecting cells with near-native CENP-B expression levels (Figure 7A). This is important, as the expression level had a distinct influence on observed CENP-B dynamics: FRAP experiments in highly-expressing cells revealed highly augmented CENP-B recovery kinetics compared to CENP-B at native levels (Figure S10E).”

2. The authors state their CENP-B FRAP results are consistent with a previous study (Hemmerich et al., 2008). In the study, Hemmerich et al. measured CENP-B mobility in detail: short-term and long term FRAP experiments in various cell cycle stages. The authors' FRAP results look consistent to the short-term FRAP in G1/S-phase cells in Hemmerich et al., e.g ~20% immobile fraction. However, this immobile fraction is in fact mobile in long-term FRAP experiments as slow mobile population. Hemmerich et al concluded that most of CENP-B in G1/S-phase cells are mobile but separated into two populations: slow-exchange and fast-exchange populations, and suggested that these different resident times were potentially caused by different DNA-binding modes: one directly binds to B-boxes with high-affinity and other binds to adjacent centromeric DNA. Considering this, to strengthen the authors' conclusion, the long-term FRAP experiments (data for the slow mobile fractions) should be important.

In addition, more than 80% of CENP-B population is immobile in G2-phase cells. The authors should specify the cell cycles stages of cells measured in their FRAP assay.

In our new experiments we now arrest cells in G1 to ensure that the measurements are performed in the same cell cycle stage (Supplementary Fig. S10C). Together with controlling expression levels, we now only observe the slow process reported in Hemmerich et al. Only when overexpressing CENP-B at higher levels (similar to Hemmerich et al), the faster process becomes apparent, even in G1 (Supplementary Fig. S10E). We thus conclude, that in our system, under controlled low CENP-B expression, CENP-B exchanges on the ~ 10 min timescale and the fast process (with a time constant of 100s of seconds) results from protein overexpression. Nevertheless, we see a distinct speed-up of CENP-B dynamics upon CENP-A depletion, albeit the FRAP recovery rate constants are now slower (Fig. 7B).

3. In the manuscript, CENP-A was knocked-down using siRNAs and the CENP-A levels were examined by immunoblots. But even though CENP-A is knocked-down or -out, CENP-A protein still remains on centromeres for several days (see Fachinetti et al., 2013, about 50% of CENP-A remains on centromeres at 48 h after CENP-A knock-out). The authors should examine CENP-A levels on the centromeres to show how much CENP-A is depleted from “centromeres” in their experiments.

We agree and have now redone the experiments using AID system to rapidly degrade CENP-A. We confirmed the depletion of CENP-A from centromeres before the FRAP experiments, i.e. the CENP-A levels are under the detection limit at the centromeres (Supplemental Fig.S10A).

4. Fachinetti et al also showed CENP-B levels are unchanged at 48 h after CENP-A knock-out. Although no protein reduction does not mean no change in dynamics, the authors should also examine CENP-B levels on centromeres after CENP-A knock-down by immunofluorescence.

As we are now no longer performing a long-term CENP-A knockdown over 48 h, but rather a rapid AID-tag dependent depletion (4 h). Thus, we do not expect that overall CENP-B levels are significantly altered. However, we quantitatively determined CENP-B levels at centromeres after CENP-A depletion using immunofluorescence, observing a $\sim 24\%$ reduction in binding levels (Supplemental Fig. S10B). This change is unlikely to impact our conclusion that CENP-B exhibits faster dynamics in the absence of CENP-A.

5. During cell cycle, new CENP-A proteins are deposited to centromere in early G1, but not in S-phase (Black et al., 2007). It means that CENP-A protein levels on a centromere are reduced into half after DNA replication. Hemmerich et al showed that CENP-B is mobile in G1/S-phase and become immobile in G2-phase, indicating that CENP-B would associate to centromeric chromatin more stably when CENP-A levels at centromeres are reduced. This sounds inconsistent with the authors' conclusion. The difference could be possibly due to difference of cell cycle stages or experimental settings such as protein expression levels. I would highly recommend the authors more FRAP experiments with better controls and setup to prove their model based on in vitro single-molecular FRET in the native context. It will improve and strengthen the manuscript.

A possible explanation is that the interplay between CENP-A and CENP-B is most important during centromere establishment in G1/S. Hemmerich et al. state that : 'in G2 and M phase, the majority of CENP-B is stably incorporated into the centromere complex'. At this point, CENP-B is no longer mobile and dynamic ($\tau \sim 50$ min), and centro-chromatin is solidly established, so any chromatin-dependent effects are no longer observed.

As mentioned in the reply to point 2., we have now constrained our analysis to G1 via a G1 arrest, and performed the FRAP experiments using much better controlled conditions (cell cycle, expression levels, CENP-A AID depletion etc.). With our new workflow we can now confirm a clear dependence of CENP-B dynamics on CENP-A presence.

6. Fig1F,H. how much of CENP-A fibers observed did show FRET? Some fibers should form the stacked-nucleosome structure as shown in the model in figures. But others maybe not.

We quantified the number of traces showing FRET in Figure 2C and in Table S4. We added the information to the figure legend.

And also in Fig1H, what portion of traces did show E fret average > 0.2? This 0.2 form intermediated E fret defined in Fig2? More explanation for its criteria would be helpful.

We 'gated' the FRET to pick up most of the broad, intermediate FRET population, while not including the peak of the low FRET population (see plot taken from Figure 2A).

We added an explanation about the selection criteria to the Figure legend/SI text.

"Traces were gated for $E_{FRET} > 0.2$, avoiding the population of zero- E_{FRET} traces (see also histograms in Figure 2A)."

Fig1H. although it is published, it would be nice to have the donor-acceptor cross-correlation analysis results of H3 control of the current dataset.

We agree and have included it into the figure (new panel Fig. 1G).

Fig2A left. the trace seems to have high ERET after 60 s, but no E fret is shown in the bottom graph. I might miss but it would be helpful if there are some explanations. The right graphs need y axis definition.

We thank the reviewer for catching the mistake, we have corrected the figure. Moreover, we have added a y-axis and definition to the right graph (rel. population), also in Figs. 4 and 5.

Fig2A. does "40 mM KCl" in the text mean "no Mg^{2+} (or 0 mM Mg^{2+})" in the figure? If so, the authors might want to use consistent terms through the manuscript.

Indeed, 40 mM KCl is present in all samples, only the concentration of Mg^{2+} is changed. We adapted the text to 0 mM Mg^{2+} consistently in the manuscript.

Fig2A,B. are the high-FRET population centered at E fret values in the figures correct? I wanted to just make sure, because the values are different from those in the text.

Due to several refits, the figures, text and Table S4 were slightly divergent. This has been corrected.

Fig2C,F. It is unclear to me what is "the large-scale fluctuations". More information about how authors determined the Dynamic traces (as well as high and low FRET traces) is helpful.

We added a more explicit definition into the SI:

"Traces were finally sorted into 'static' and 'dynamic' dependent on the existence of anticorrelated intensity fluctuations in donor- and acceptor channels. Low and high-FRET populations were assigned by the Gaussian distributions obtained after fitting to the formula above (Low FRET: $< 0.2 E_{FRET}$ center of the distribution, High FRET: $> 0.2 E_{FRET}$ center of the distribution)."

Fig3F. in the text, short and long resident times are $t_{off,1}$ and 2, respectively. But in table S5 they are $t_{off,0}$ and 1. They would be 1 and 2 (or 0 and 1 in the text). The authors should use consistent terms through the manuscript. I found several other inconsistencies. The authors might want to check through the manuscript.

We checked the manuscript and removed the inconsistencies, in particular with regards to $\tau_{off,1}$ and $\tau_{off,2}$

Fig3I. in the text authors say that “not significantly different compared to those for naked DNA (Figures 3I)”. But no statistical analysis between naked DNA and H3 MN there. Authors should revise the text or add statistical analysis in the figure.

We revised the text to:
“similar to those for naked DNA (Figures 3I)”

p5, l.2, “...this resulted in stable E fret ~ 0.5 over several tens of seconds [39].” Here, probably, the authors should cite their own results such as Fig2B (or add control H3 fiber results in Fig1 and cite it) instead of ref39. It is an important control to show the experimental condition used in the work is fine as previously reported.

We added a trace showing stable FRET for H3 chromatin in Figure 1, and revised the text accordingly.

p7, l.8 “In such an experiment, individual naked DNA molecules, reconstituted nucleosomes or chromatin fibers, all labeled...(Figure 3A)” Figure 3 showed results with mono-nucleosomes. It should be “In such an experiment, individual naked DNA molecules or reconstituted nucleosomes, all labeled...(Figure 3A)”

We corrected the mistake.

p13, l8. “siRNA against CENP-B” should be “siRNA against CENP-A”

We redid those experiments, avoiding the use of siRNA

Reviewer #3 (Remarks to the Author):

Major concerns:

1. Cell based-experiment: Fig. S8A shows massive overexpression of the CENP-B relative to the endogenous protein. It doesn't make sense to do the experiment with unnaturally high levels of CENP-B protein around, and I don't think the results are relevant to physiological understanding at centromeres, in their current form. I'm not sure gene replacement and/or inducible expression is necessary, but lower expression is necessary for this experiment to be relevant to what might be happening at natural centromeres.

We agree with the reviewer. We have now generated stable cell lines and finely tuned mCherry-CENP-B expression to arrive at near-native levels as judged by immunofluorescence and mCherry emission. See also our response to reviewer 2, point 1.

2. Cell based-experiment: It makes no sense to use siRNA to knock down CENP-A. It has been known to complicate interpretations in functional experiments and requires a wind down (so with complete removal of CENP-A message, the protein is down to 25% after 2 cell divisions [i.e. at the 48 h timepoint of their experiment]). The centromere is slowly being affected during all of this and so interpretations

of their data is further compromised. Inducible degradation of CENP-A is the standard for almost a decade since introduced by the Cleveland lab. It permits a clear way to test the effect of CENP-A removal within a few hours, or sooner, from the initiation of depletion. The experiments need to be performed in this condition.

We agree and have used a cell line, received from the Fachinetti lab, that allows rapid depletion of CENP-A via the AID tag. We engineered these cells to also express mCherry-CENP-B under an inducible promoter. Together, this allowed us to perform FRAP under controlled conditions (cell cycle, CENP-B levels) after rapid CENP-A depletion (4 h after IAA addition). See new Fig. 7 and Supplemental Fig. 9 for details and new data. In short, with these cells and our new workflow we can now confirm a clear dependence of CENP-B dynamics on CENP-A presence.

Minor Concerns:

3. The introduction says that the 'RG-Loop' is important for CENP-N and CENP-C recruitment. For CENP-N this is now considered controversial with recent structures from the Barford and Musacchio labs. It would be helpful to mention the studies in support or denial of this mode of binding. Also, I am not aware of the data that this loop is important for CENP-C, so that should certainly be cited there.

We corrected the sentence accordingly, removed the mention of CENP-C, added references for CENP-N and discussed the recent structures.

4. CENP-B is referred to as a 'kinetochore' component. Unlike CENP-A, it is also found at human centromeres at an adjacent region of the chromosome that does not build a kinetochore. Also unlike CENP-A and other components of the centromere, CENP-B is dispensable for kinetochore formation. Maybe 'component of centromere/centromeric chromatin' is better?

We changed all instances where we were referring to CENP-B as a kinetochore protein to 'component of centromeric chromatin'.

5. When referring to "CENP-A chromatin", are these nucleosome arrays containing all CENP-A or only the middle tetramers are "CENP-A chromatin"? Please clarify in the text.

The arrays contain all CENP-A. We clarified this now.

6. In figure 2A-B, the E FRET values in the text (page 5) and the figure do not match. The text says "... a high-FRET population centered at E FRET values of 0.44 for CENP-A and 0.54 for H3 chromatin (Figure 2A,B)" however the figure marks the E FRET peak at 0.42 and 0.56, for CENP-A and H3 respectively.

We corrected all the values.

REVIEWERS' COMMENTS

Reviewer #1 (Remarks to the Author):

My major concern was about the biological relevance of the chromatin model: a 12-mer of CENP-A nucleosome on a non alpha-satellite DNA, with only 1 or 2 CENP-B boxes. In the revised manuscript the authors discuss more precisely their model and its implications/limitations. Moreover, the authors performed an important additional CryoEM experiment on CENP-A nucleosome in the presence of CENP-B. In this new experiment alpha-satellite DNA is now used, representing a centromeric mononucleosome in a better way. The CryoEM data showed that CENP-B binding indeed increases the flexibility of alpha-satellite DNA ends in the context of CENP-A nucleosome, confirming smFRET results and clarifying a possible ambiguity in the smFRET interpretation.

I believe the manuscript can be now accepted for publication.

Reviewer #3 (Remarks to the Author):

The authors made substantial additions, even including cryo-EM that supports their overall findings. New experiments were included that effectively address the major comments from the initial round of review. I focused on my initial comments and those of Reviewer #2. For all of those comments/concerns, I feel that the authors satisfactorily addressed them. My recommendation is to publish the paper as it now stands.